# In-Context Adaptation to Concept Drift for Learned Database Operations

Jiaqi Zhu [1]   Shaofeng Cai [* 2]   Yanyan Shen [3]   Gang Chen [4]   Fang Deng [1]   Beng Chin Ooi [2 4]

## Abstract

Machine learning has demonstrated transformative potential for database operations, such as query optimization and in-database data analytics. However, dynamic database environments, characterized by frequent updates and evolving data distributions, introduce *concept drift*, which leads to performance degradation for learned models and limits their practical applicability. Addressing this challenge requires efficient frameworks capable of adapting to shifting concepts while minimizing the overhead of retraining or fine-tuning.

In this paper, we propose FLAIR, an online adaptation framework that introduces a new paradigm called *in-context adaptation* for learned database operations. FLAIR leverages the inherent property of data systems, i.e., immediate availability of execution results for predictions, to enable dynamic context construction. By formalizing adaptation as $f : (\mathbf{x} \,|\, \mathcal{C}_t) \to \mathbf{y}$, with $\mathcal{C}_t$ representing a *dynamic context memory*, FLAIR delivers predictions aligned with the current concept, eliminating the need for runtime parameter optimization. To achieve this, FLAIR integrates two key modules: a Task Featurization Module for encoding task-specific features into standardized representations, and a Dynamic Decision Engine, pre-trained via Bayesian meta-training, to adapt seamlessly using contextual information at runtime. Extensive experiments across key database tasks demonstrate that FLAIR outperforms state-of-the-art baselines, achieving up to $5.2\times$ faster adaptation and reducing error by 22.5% for cardinality estimation.

[1]Beijing Institute of Technology, Beijing, China [2]National University of Singapore, Singapore [3]Shanghai Jiao Tong University, Shanghai, China [4]Zhejiang University, Hangzhou, China. Correspondence to: Shaofeng Cai <shaofeng@comp.nus.edu.sg>.

*Proceedings of the $42^{st}$ International Conference on Machine Learning*, Vancouver, Canada. PMLR 267, 2025. Copyright 2025 by the author(s).

## 1. Introduction

Data systems are increasingly integrating machine learning functionalities to enhance performance and usability, marking a paradigm shift in how data is managed and processed in databases (Ooi et al., 2024; McGregor, 2021; Li et al., 2021; Ooi et al., 2015). The integration has transformed key database operations such as query optimization, indexing, and workload forecasting into more precise, efficient, and adaptive processes (Zhang et al., 2024a; Kurmanji & Triantafillou, 2023; Anneser et al., 2023).

Despite these advancements, learned database operations face a persistent challenge: *concept drift*. Databases are inherently dynamic, undergoing frequent insert, delete, and update operations that result in shifts in data distributions and evolving input-output relationships over time (Zeighami & Shahabi, 2024). These drifts, often subtle but cumulative, can alter the patterns and mappings that traditional machine learning models rely upon, rendering their assumptions of static distributions invalid. This phenomenon requires adaptive methods for maintaining predictive accuracy in dynamic database environments.

Traditional *reactive training-based adaptation* approaches to handling concept drift, such as transfer learning (Jain et al., 2023; Kurmanji & Triantafillou, 2023; Kurmanji et al., 2024), active learning (Ma et al., 2020; Li et al., 2022), and multi-task learning (Kollias et al., 2024; Wu et al., 2021), come with significant drawbacks in learned database operations. As illustrated in Figure 1, delays and costs in post-deployment data collection and model updates, and reliance on static mappings limit their practicality in dynamic database environments (Kurmanji et al., 2024; Li et al., 2022). In addition, they process each input independently. The negligence of inter-query dependencies and shared contextual information in databases results in poor modeling of database operations. Addressing these limitations raises two critical challenges: (1) *How can we support one-the-fly adaptation to constantly evolving data without incurring the overhead of frequent retraining or fine-tuning in databases?* (2) *How can we dynamically inject contextual information into the modeling process to achieve context-aware prediction for learned database operations?*

To address these challenges, we introduce FLAIR, an eFficient and effective onLine AdaptaIon fRamework that

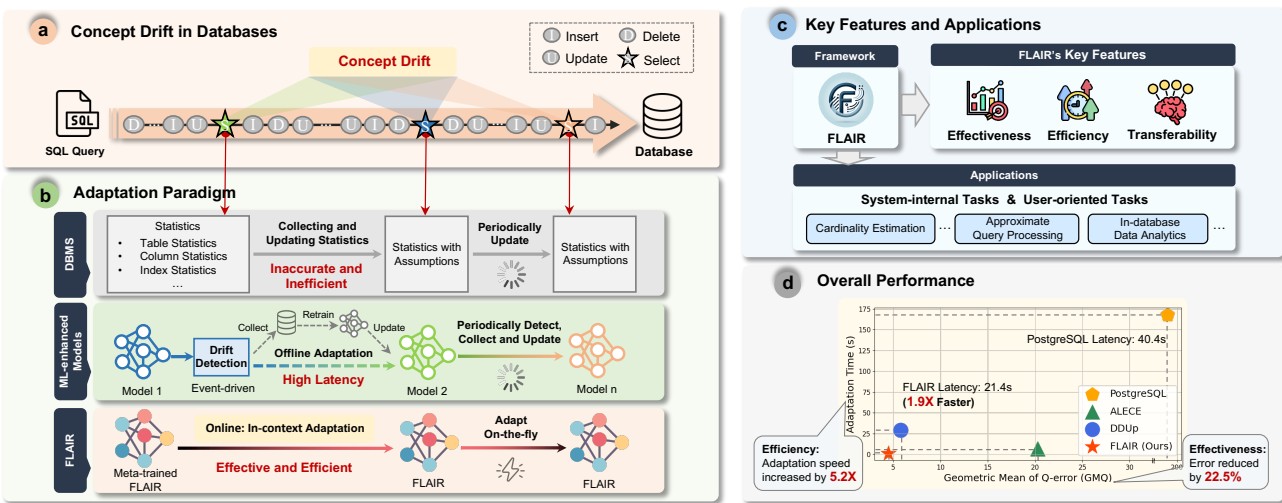

Figure 1: FLAIR in a nutshell. (a) An example of concept drifts in a dynamic database setting. (b) Adaptation paradigm for handling concept drifts in FLAIR and two conventional approaches. (c) Key features and applications of FLAIR. (d) A preview comparison of FLAIR against PostgreSQL and state-of-the-art approaches for handling dynamic databases.

establishes a new adaptation paradigm for learned database operations. FLAIR is built on a unique property of database operations: the immediate availability of execution results for predictions in the database. These results, serving as ground-truth labels, provide real-time feedback that enables seamless adaptation. FLAIR leverages this property to dynamically adapt to evolving concepts using such *contextual cues* from databases. Formally, FLAIR models the mapping as $f : (\mathbf{x} \,|\, \mathcal{C}_t) \rightarrow \mathbf{y}$, where $\mathbf{x}$ denotes the input query, $\mathcal{C}_t$ is the current context consisting of recent pairs of queries and their execution results, and $\mathbf{y}$ is the predicted output.

To achieve *in-context adaptation* for learned database operations, FLAIR introduces two cascaded modules: the *task featurization module* (TFM) and the *dynamic decision engine* (DDE). The TFM encodes database operations into standardized task representations, extracting informative features and producing a unified, structured input format. This ensures consistency and efficiency across diverse tasks within databases. The *dynamic decision engine* functions as the core of FLAIR, delivering predictions that can adapt to evolving concepts. To this end, we introduce a Bayesian meta-training mechanism that utilizes synthetic prior distributions to pretrain FLAIR with a comprehensive knowledge base, pre-adapting it to handle diverse and dynamic scenarios. Unlike traditional reactive approaches, FLAIR eliminates the need for compute-intensive parameter optimization after deployment. To the best of our knowledge, FLAIR is the first framework to enable on-the-fly and context-aware adaptation in dynamic data systems.

We summarize our main contributions as follows:

- We propose a novel in-context adaptation framework FLAIR, designed to address the persistent challenge of concept drift in dynamic data systems with high effi-

ciency and effectiveness.

- FLAIR introduces Bayesian meta-training that enables robust and transferable learning from dynamic distributions, thus eliminating the need for costly parameter retraining or fine-tuning after deployment.

- FLAIR is designed as a task-agnostic framework that enhances a wide range of learned database operations. These include system-internal tasks such as cardinality estimation, and user-oriented applications like approximate query processing and in-database data analytics.

- Extensive experiments show FLAIR's superior performance in dynamic databases, achieving a $5.2\times$ speedup in adaptation and a 22.5% reduction in GMQ error for cardinality estimation. Furthermore, by integrating FLAIR with PostgreSQL, we achieve up to a $1.9\times$ improvement in query execution efficiency.

## 2. Preliminaries

**Problem Formulation.** Consider a database $\mathbb{D}$ consisting of a set of relations (tables) $\{\mathbf{R_1}, ..., \mathbf{R_N}\}$. Each relation $\mathbf{R_i}$ has $n_i$ attribute fields (columns), $\mathbf{R_i} = (\mathbf{a_1^i}, ..., \mathbf{a_{n_i}^i})$, where the attributes correspond to either categorical or numerical features in prediction. In this paper, we focus on select-project-join (SPJ) queries executed alongside a mix of insert, delete, and update operations. The challenge addressed is *concept drift*, an intrinsic property of databases, described as a shift in the relationship between queries and their corresponding predictive outputs over time.

**Definition 2.1** (Concept Drift in Databases)**.** Let $\mathbf{d}_t$ be the underlying data of a database at time $t$, and $\mathbf{q}_t$ denote a user query at time $t$. Given the data-query pair $(\mathbf{d}_t, \mathbf{q}_t)$, let $\mathbf{y}_t$ be the corresponding prediction output (e.g., row counts in

cardinality estimation). Concept drift occurs at time $t$ if the joint distribution of queries, data, and predictions changes, i.e., $P_t(\mathbf{q}, \mathbf{d}, \mathbf{y}) \neq P_{t+1}(\mathbf{q}, \mathbf{d}, \mathbf{y})$.

Here, concept drift is driven by two interrelated sources: (1) Query drift, from evolving user behavior. (2) Data drift, caused by insert/delete/update operations that change underlying data distributions. Notably, changing data not only changes the marginal distribution $P(\mathbf{d})$, but also the conditional distribution $P(\mathbf{y}|\mathbf{q}, \mathbf{d})$. That is, the same queries may yield different outputs over time. This suggests that concept drift in databases involves shifts in the joint distribution of queries, data, and predictions, and their interaction. For example, in an e-commerce database, incremental updates, such as new product additions, customer preference changes, or promotional campaigns, can lead to significant concept drift in product recommendations.

**Learned Database Operations.** Learned database operations employ machine learning models to enhance specific tasks in databases, such as cardinality estimation and approximate query processing. Let $\mathcal{M}_D(\cdot; \Theta)$ denote a prediction model parameterized by $\Theta$ in a database D. $\mathcal{M}_D(\mathbf{x}; \Theta)$ takes a query $\mathbf{x}$ as input and makes a prediction, e.g., the number of rows matching $\mathbf{x}$ for cardinality estimation.

However, a model becomes stale when concept drift occurs. Formally, the model $\mathcal{M}_{D_t}(\mathbf{x}; \Theta_t)$ trained on data $D_t$ becomes ineffective at time $t + \Delta t$, if $P_t(\mathbf{x}, \mathbf{y}) \neq P_{t+\Delta t}(\mathbf{x}, \mathbf{y})$. Traditional approaches require periodic data recollection and model retraining to maintain accuracy. This incurs high costs. Our objective is to ensure that the model $\mathcal{M}_{D_t}(\mathbf{x}; \Theta_t)$ can be efficiently and effectively adapted to evolving data distributions with these resource-intensive processes in database environments.

**In-context Learning with Foundation Models.** Foundation models have seen rapid advancements in capability and scope (Radford et al., 2019; Raffel et al., 2020; Brown et al., 2020; Achiam et al., 2023), which give rise to a transformative paradigm called *in-context learning* (ICL). ICL embeds context into the model input, and leverages foundation models' broad learned representations to make predictions based on limited contextual examples, thus bypassing the need for parameter updates after deployment. This paradigm drastically cuts compute demands and facilitates various applications (Sun et al., 2022; Dong et al., 2022). A notable application for tabular data is *Prior-data Fitted Networks* (PFNs) (Müller et al., 2022; Hollmann et al., 2023; Helli et al., 2024), which are pre-trained on synthetic datasets sampled from pre-defined priors. This enables PFNs to pre-adapt to dynamic environments by effectively modeling uncertainties and various distributions, making PFNs suitable for scenarios with frequent updates and concept drift. Please refer to Appendix B for more details on ICL and PFNs. In this paper, we aim to utilize real-time feedback

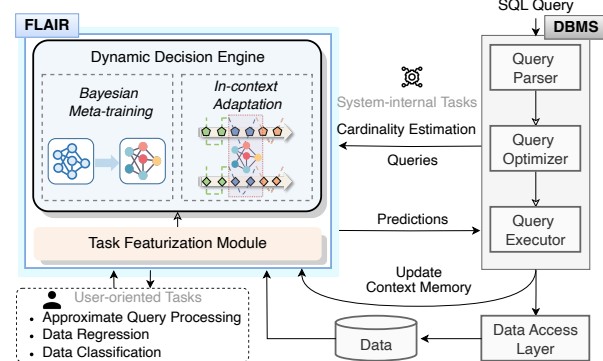

Figure 2: FLAIR for dynamic data systems.

from database environments and explore how to support in-context adaption for learned database operations.

# 3. FLAIR for Dynamic Data Systems

As illustrated in Figure 2, FLAIR introduces a dual-module architecture that addresses concept drift in dynamic databases. First, to provide a unified interface across different tasks, the Task Featurization Module (TFM) extracts task-specific features from database operation for the subsequent modeling. Second, the Dynamic Decision Engine (DDE) is pre-trained via Bayesian meta-training on dynamic distributions of tasks, pre-adapting it to diverse tasks encountered during inference. After meta-training, DDE utilizes the real-time feedback from databases as the latest contextual information to dynamically adapt to the current task. The workflow of FLAIR $\mathcal{M}_F$ is outlined as:

$$\mathcal{M}_F(\mathbf{x}; \Theta_{\mathcal{T}}, \Theta_{\mathcal{D}}) = \mathcal{M}_{DDE}(\mathcal{M}_{TFM}(\mathbf{x}; \Theta_{\mathcal{T}}); \Theta_{\mathcal{D}}), \quad (1)$$

which comprises two cascading modules, the TFM $\mathcal{M}_{TFM}$ and the DDE $\mathcal{M}_{DDE}$ parameterized by $\Theta_{\mathcal{D}}$ and $\Theta_{\mathcal{T}}$, respectively. We introduce the technical details below.

## 3.1. Task Featurization Module

The TFM is designed to standardize database operations into structured inputs for downstream modeling. It first encodes data and queries of database operations into *data vectors* and a *query vector* respectively, and then extracts a *task vector* via cross-attention that integrates their interactions.

### 3.1.1. DATA AND QUERY ENCODING

**Data Encoding.** Each attribute (i.e., column) in the database is represented as a histogram, which captures its distribution. Formally, for an attribute $\mathbf{a_n^i}$ in relation $\mathbf{R_i}$, the histogram $\mathbf{x}_n^i = [x_1, \cdots, x_\delta]$ uses $\delta$ bins to discretize the range of the attribute. After scaling to $[0, 1]$, these histograms are aggregated to form comprehensive data vectors $\mathrm{X_D}$ of dimension $\delta \times \sum_{i=1}^{N} n_i$, where $N$ is the total number of relations, and $n_i$ is the number of attributes in relation $\mathbf{R_i}$.

**Query Encoding.** Queries are represented as vectors capturing structural and conditional information. Join predicates, e.g., $\mathbf{R_i a_{n_i}^i} = \mathbf{R_j a_{n_j}^j}$, are encoded into binary vectors $\mathbf{q}_J$ via one-hot encoding, while filter predicates, e.g., $\mathbf{R_i a_{n_i}^i}$ op $\mho$ with op $\in \{<, \leqslant, \geqslant, >, =\}$ being the comparison operators and $\mho$ the condition value, are encoded into boundary vectors $\mathbf{q}_F$. For details of encoding schemes for these predicates, please refer to Appendix C. The final *query vector* $\mathbf{q}_{\mathcal{Q}} = <\mathbf{q}_J, \mathbf{q}_F>$ concatenates these encodings.

### 3.1.2. TASK FEATURIZATION

To derive the task vector, we adopt a lightweight transformer (Vaswani et al., 2017) architecture following (Li et al., 2023b), which employs hybrid attention mechanisms to extract deep latent features. The task featurization process starts with a *data modeling phase,* where data vectors $\mathrm{X_D}$ are processed through a series of Multi-head Self-attention (MHSA) layers, interleaved with Feed-forward Network (FFN), Layer Normalization (LN), and residual connections. This is to capture implicit joint distributions and complex dependencies among attributes within $\mathrm{X_D}$:

$$\hat{\mathbf{Z}}^l = \mathrm{MHSA}(\mathrm{LN}(\mathbf{Z}^{l-1})) + \mathbf{Z}^{l-1} \quad (2)$$

$$\mathbf{Z}^l = \mathrm{FFN}(\mathrm{LN}(\hat{\mathbf{Z}}^l)) + \hat{\mathbf{Z}}^l \quad (3)$$

where MHSA operations are formulated as:

$$\mathbf{Q}^{l,m} = \mathbf{Z}^{l-1}\mathbf{W}_q^{l,m}, \mathbf{K}^{l,m} = \mathbf{Z}^{l-1}\mathbf{W}_k^{l,m}, \mathbf{V}^{l,m} = \mathbf{Z}^{l-1}\mathbf{W}_v^{l,m} \quad (4)$$

$$\mathbf{Z}^{l,m} = \mathrm{softmax}(\frac{\mathbf{Q}^{l,m}(\mathbf{K}^{l,m})^T}{\sqrt{d_k}})\mathbf{V}^{l,m}, \; m = 1, \cdots, M \quad (5)$$

$$\mathbf{Z}^l = \mathrm{concat}(\mathbf{Z}^{l,1}, \cdots, \mathbf{Z}^{l,M})\mathbf{W}_o^l \quad (6)$$

where $\mathbf{Z}^0$ is composed of data vectors from $\mathrm{X_D}$, and $M$ is the number of attention heads. $\mathbf{Q}^{l,m}$, $\mathbf{K}^{l,m}$, and $\mathbf{V}^{l,m}$ denote the *query*, *key*, and *value* of the $m$-th head in the $l$-th layer, obtained via transformation matrices $\mathbf{W}_q^{l,m}$, $\mathbf{W}_k^{l,m}$, and $\mathbf{W}_v^{l,m}$, respectively. $\mathbf{Z}^l$ is the output of the $l$-th layer, and $\mathbf{W}_o^l$ is the output transformation matrix.

In the subsequent *interaction modeling phase*, the output of the data modeling phase $\mathbf{Z}_{\mathcal{O}}$ is further refined via the Multi-head Cross-attention (MHCA) mechanism. Unlike MHSA, $\mathbf{Z}_{\mathcal{O}}$ serves dual roles as both the *keys* and *values*, while the query vector $\mathbf{q}_{\mathcal{Q}}$ acts as the *query* in MHCA. The query vector $\mathbf{q}_{\mathcal{Q}}$ interacts with every vector in $\mathbf{Z}_{\mathcal{O}}$ through key and value transformations, allowing TFM to dynamically focus on the features in $\mathbf{Z}_{\mathcal{O}}$ pertinent to the query. For each attention head $m$ in MHCA, we have:

$$\mathbf{z}^m = \mathrm{softmax}(\frac{\mathbf{q}_{\mathcal{Q}}(\mathbf{Z}_{\mathcal{O}}\mathbf{W}_k^m)^T}{\sqrt{d_k}})(\mathbf{Z}_{\mathcal{O}}\mathbf{W}_v^m). \quad (7)$$

The final task vector $\mathbf{z}_{\mathcal{T}}$ is obtained by further processing the MHCA output through an FFN layer followed by LN with residual connections. In this way, the task vector $\mathbf{z}_{\mathcal{T}}$ contains task-specific information of both data attribute relations and query conditions, providing comprehensive task representations for the subsequent modeling in the DDE.

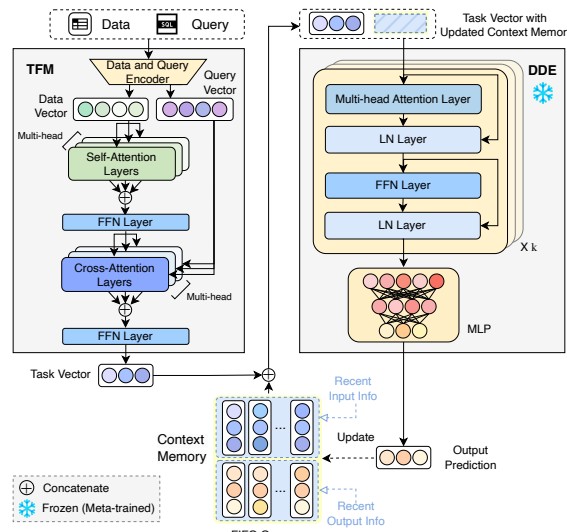

Figure 3: The architecture of FLAIR.

### 3.2. Dynamic Decision Engine

The DDE forms the core module of FLAIR. As illustrated in Figure 3, the DDE takes the task vector prepared by the TFM to provide real-time, context-aware predictions across various tasks. It comprises two phases: *Bayesian meta-training* and *in-context adaptation*.

### 3.2.1. BAYESIAN META-TRAINING

DDE is pre-trained using synthetic datasets sampled from prior distributions, which equips the model with broad generalization capabilities, enabling rapid adaptation to unseen tasks. The meta-training is based on Bayesian inference theory. Formally, for a given sample $\mathbf{x}$ with the evolving concept represented by a set of $c$ observed sample pairs $\mathcal{C} = \{(\mathbf{y}_i, \mathbf{x}_i)\}_{i=1}^c$ from the current task, the Posterior Predictive Distribution (PPD) of task predictive modeling is:

$$p(\mathbf{y}|\mathbf{x}, \mathcal{C}) = \int_\Phi p(\mathbf{y}|\mathbf{x}, \phi)p(\phi|\mathcal{C})d\phi \quad (8)$$

$$\propto \int_\Phi p(\mathbf{y}|\mathbf{x}, \phi)p(\mathcal{C}|\phi)p(\phi)d\phi \quad (9)$$

where the task distribution $p(\phi)$ is sampled from curated prior distributions $\Phi$ to diversify the adaptability of DDE to different prediction tasks. Notably, to capture complex dependencies and uncover underlying causal mechanisms, we employ Bayesian Neural Networks (BNNs) (Neal, 2012) and Structural Causal Models (SCMs) (Pearl, 2009; Peters et al., 2017) in constructing the prior distribution following PFNs (Hollmann et al., 2023).

Based on the PPD formulation in Eq. (9), we first generate synthetic datasets, namely the concept $\mathcal{C}$ of observed samples from the task distribution $p(\phi)$, i.e., $\mathcal{C} \sim p(\mathcal{C}|\phi)$. Second, we sample the data points $(\mathbf{x}, \mathbf{y})$ for predictive modeling from $p(\mathbf{x}, \mathbf{y}|\phi)$. Next, we can train DDE using

the input-output configuration via the loss:

$$\mathcal{L}_{DDE} = \mathbb{E}_{((\mathbf{x},\mathcal{C}),\mathbf{y})\in p(\phi)}[-\log q_\theta(\mathbf{y}|\mathbf{x},\mathcal{C})] \quad (10)$$

where the $q_\theta(\mathbf{y}|\mathbf{x},\mathcal{C})$ is the model's predictive distribution parameterized by $\theta$. By minimizing this expected negative log probability $\mathcal{L}_{DDE}$, DDE is trained to maximize the likelihood of the observed data under the current task distribution $p(\phi)$. In particular, $\mathcal{L}_{DDE}$ can be formalized as follows for different types of tasks, corresponding to regression and classification tasks, respectively.

$$\mathcal{L}_{reg} = \mathbb{E}_{((\mathbf{x},\mathcal{C}),\mathbf{y})\in p(\phi)}\left[\frac{(\mathbf{y}-\mu)^2}{2\sigma^2} + \log\sigma\right] \quad (11)$$

$$\mathcal{L}_{cls} = \mathbb{E}_{((\mathbf{x},\mathcal{C}),\mathbf{y})\in p(\phi)}\left[-\sum_{k=1}^{K}\mathbb{I}_{\mathbf{y}=k}\log q_\theta(\mathbf{y}=k|\mathbf{x},\mathcal{C})\right] \quad (12)$$

where $\mu$ and $\sigma$ are the mean and standard deviation in regression tasks, $\mathbb{I}(\cdot)$ is the indicator function and $q_\theta(\mathbf{y}=k|\mathbf{x},\mathcal{C})$ is the predicted probability of class $k$ in classification tasks.

**Remark.** We note that the Bayesian meta-training is performed only once on the curated prior distributions across various tasks. With Bayesian meta-training, FLAIR is enabled to quickly adapt to new concepts using a limited set of observed samples of the concept. This offers several advantages: (1) *Cost-effective Data Collection*: Generating synthetic data is significantly more cost-effective and faster than traditional data collection. (2) *One-time Effort*: The process is a one-time effort, eliminating frequent retraining after deployment. (3) *No Privacy Issues*: Synthetic data does not contain real user information, thereby circumventing privacy and security concerns. (4) *Scalability*: This strategy allows for easy adoption of desired prior task distributions instead of rebuilding the entire model from scratch.

### 3.2.2. IN-CONTEXT ADAPTATION

During inference, we query the meta-trained DDE with the tuple $(\mathbf{z}_\mathcal{T},\mathcal{C})$ as input, where $\mathcal{C} = (\mathcal{Q}_{pmt},\mathcal{Y}_{pmt})$, termed as *context memory*, contains contextual information of the current task. $\mathcal{Q}_{pmt}$ and $\mathcal{Y}_{pmt}$ denote the sequences of recent queries and the system feedback, namely true outputs, which are organized into two separate first-in, first-out (FIFO) queues of size $\varrho$. This strategy enables DDE to dynamically adapt to new concepts guided by the context memory during inference, thus avoiding backpropagation-based adaptation such as fine-tuning or retraining.

**Remark.** To better understand the in-context adaptation mechanism, we examine the key differences between FLAIR and existing learned approaches. Existing methods like Marcus et al. (2021); Zhao et al. (2022); Wang et al. (2023a) typically learn a static mapping from input to output as in Eq. 13, which assumes a fixed data distribution. When concept drift occurs in the time interval $\Delta t = t' - t$, i.e., $\mathcal{D}_t \neq \mathcal{D}_{t'}$ and $P_t(\mathbf{x},\mathbf{y}) \neq P_{t'}(\mathbf{x},\mathbf{y})$, the mapping $f_{\mathcal{D}_t,\Theta_t}$ from the input to the output should change accordingly. To

handle concept drift, these methods require collecting sufficient samples from the new distribution and updating the mapping $f_{\mathcal{D}_t,\Theta_t}$ with parameter $\Theta_t$ based on these samples, so as to obtain a new mapping function $f_{\mathcal{D}_{t'},\Theta_{t'}}$ with parameter $\Theta'_t$ that aligns with the new distribution $\mathcal{D}_{t'}$. In contrast, our new paradigm essentially learns a *conditional mapping* as formulated in Eq. 14, which explicitly models the evolving concept provided by the context memory $\mathcal{C}_t$ as the context of the current distribution $\mathcal{D}_t$.

$$\forall t, \ f_{\mathcal{D}_t,\Theta_t} : \mathbf{x} \to \mathbf{y} \quad (13)$$

$$\forall t, \ f_{\mathcal{D}_t,\Theta} : (\mathbf{x}\,|\,\mathcal{C}_t) \to \mathbf{y} \quad (14)$$

This adaptability via the in-context adaptation mechanism is well-suited for databases. When a query is executed, the corresponding system output becomes immediately available and can be stored in the context memory to provide supervision for contextualized predictions of subsequent queries. Also, For user-oriented tasks like data classification, the context memory within FLAIR allows for online user feedback, which facilitates the development of a customized system better aligned with user preferences.

### 3.3. FLAIR Workflow: Training to Inference

**Training.** FLAIR is trained in two stages: (i) First, the $\mathcal{M}_{DDE}$ module undergoes a *one-off meta-training* phase using $\mathcal{L}_{DDE}$ in Eq. 10 across crafted task distributions. Note that the meta-training is not to optimize FLAIR directly on end tasks but to prepare DDE to adapt to new tasks met during inference without further training. (ii) Second, the $\mathcal{M}_{TFM}$ module is trained to extract informative latent features that are critical for the specific tasks at hand. The training of TFM is tailored to optimize performance on these tasks. This employs a task-specific loss $\mathcal{L}_{TS}$ to extract informative features for the DDE module.

**Inference.** Once trained, FLAIR is ready for concurrent online inference and adaptation in a real-time environment:

$$\mathbf{x} \Rightarrow \mathcal{M}_{TFM}(\mathbf{x};\Theta_\mathcal{T}) = \mathbf{z}_\mathcal{T} \Rightarrow \mathcal{M}_{DDE}(\mathbf{z}_\mathcal{T},\mathcal{C};\Theta_\mathcal{D}) = \mathbf{y} \quad (15)$$

$$\mathbf{x} \Rightarrow \mathcal{S}_{execute}(\mathbf{x}) = \mathbf{y}^* \Rightarrow (\mathbf{z}_\mathcal{T},\mathbf{y}^*) \xrightarrow{\text{update}} \mathcal{C} \quad (16)$$

where $\mathcal{S}_{execute}(\cdot)$ is the data system executor that produces the actual system output $\mathbf{y}^*$. Fundamentally, FLAIR streamlines the model update process by replacing the traditional, cumbersome backpropagation with an efficient forward pass via meta-training and in-context adaptation mechanism.

FLAIR efficiently accommodates large dynamic databases through incremental histogram maintenance in $O(N_v)$ with $N_v$ modified records and adapts to concept drift using a FIFO key-value memory for in-context adaptation. The cross-attention mechanism operates on a single query vector and incurs only a linear overhead of $O(d_a\varrho)$, where $d_a$ is the attention dimension in DDE. This flexible and scalable workflow ensures that FLAIR learns effectively from new

tasks on-the-fly, adapting to evolving concepts in dynamic databases. Please refer to Appendix F for more discussions.

## 3.4. Model Generalization Error Bound Analysis

In this section, we analyze the generalization error bounds of FLAIR against conventional models optimized for static data, when faced with post-training data evolving. We aim to uncover the susceptibility of outdated static models to dynamic environments and showcase FLAIR's resilience. Consider a model $\hat{f}_i$ trained on dataset $D^i$ and frozen once training concludes. Subsequent $k$ single-point data operations alter the data from $D^i$ to $D^j$, where each operation is atomic, comprising either insertion or deletion[1]. $f_{D^j}$ refers to the ground-truth mapping to $D^j$. We now explore the worst-case bound on expected maximum generalization error for robustness. A proof sketch is provided below, with detailed derivations available in Appendix E.

**Theorem 3.1.** *Consider a model $\hat{f}_i$ trained on an initial dataset $D^i$, where $|D^i| = i$. After $k$ data operations, including $s$ insertion and $r$ deletion, we obtain a new dataset $D^j$ of size $|D^j| = j$, where $k = s + r > 1$ and the net difference in data size $|j - i| = |s - r|$. Suppose data in $D^j$ are i.i.d from any continuous distribution $\chi$, we have*

$$\sup_{\mathbf{x}} \mathbb{E}_{D^j \sim \chi}\big[\big|\hat{f}_i(\mathbf{x}) - f_{D^j}(\mathbf{x})\big|\big] \geqslant k - 1$$

Theorem 3.1 states that the risk of using a stale model to make predictions escalates at a minimum rate of $\Omega(k)$ as data evolves. Theoretically, to sustain a error at $\epsilon$, $\frac{\varkappa}{\epsilon+1}$ model retraining is needed for every $\varkappa$ data operation. The cost per retraining session generally involves processing the entire dataset or a significant portion thereof in the scale $\mathcal{O}(\varkappa)$ (Zeighami & Shahabi, 2024). Consequently, the amortized cost per data operation, given that retraining the model every $\epsilon + 1$ data operation, is also $\mathcal{O}(\varkappa)$. Thus, maintaining low error rates in such a dynamic setting can be computationally expensive. In contrast, our model defined as $\hat{f}(\mathbf{x}|\mathcal{C}^j)$ exhibits resilience to changes in data.

**Theorem 3.2.** *Consider FLAIR trained when the underlying database is $D^i$ and using context memory $\mathcal{C}^j$ to perform prediction when the database evolves to $D^j$, we have*

$$\sup_{\mathbf{x}} \mathbb{E}_{D^j \sim \chi}\Big[\big|\hat{f}(\mathbf{x}|\mathcal{C}^j) - f_{D^j}(\mathbf{x})\big|\Big] \leqslant \frac{\aleph}{\sqrt{\varrho}}$$

*with high probability $1-\delta$, where $\aleph = \sqrt{\frac{1}{2}(\kappa + \ln \frac{1}{\delta})} + \sqrt{\frac{\pi}{2}}$. Here, $\varrho$ is the size of the context memory $\mathcal{C}^j$, $\kappa$ is a constant reflecting the training adequacy, and data in $D^j$ is drawn i.i.d from any continuous distribution $\chi$.*

Theorem 3.2 demonstrates that the generalization error of FLAIR can be effectively controlled by the size of context

memory $\varrho$. By ensuring that $\varrho$ is sufficiently large, the generalization error remains well within the bounds of $\mathcal{O}(\frac{1}{\sqrt{\varrho}})$. Unlike traditional models that experience a linear growth in generalization error with each data operation $k$, FLAIR's error remains stable regardless of $k$, showing no performance deterioration with post-training data changes. Specifically, setting $\varrho$ to be at least $(\frac{\aleph}{k-1})^2$ ensures that the expected worst-case generalization error of FLAIR stays below static models. This aligns with existing research (Namkoong & Duchi, 2016; Sagawa et al., 2020) that considers potential distribution shifts during training bolsters model resilience after deployment. Overall, Theorem 3.2 elucidates FLAIR's theoretical superiority over static models in maintaining continuous accuracy and operational efficiency, providing a scalable solution with frequent data evolving.

# 4. Experiments

In this section, we systematically evaluate the effectiveness, efficiency, and transferability of FLAIR[2]. Extensive experiments are conducted on real-world benchmarks for cardinality estimation to test the effectiveness of FLAIR across various degrees of concept drift, followed by assessments of training and inference efficiency. We then explore FLAIR's robustness against long-term concept drift, and its transferability to representative user-oriented tasks within databases. Moreover, we integrate FLAIR with PostgreSQL to confirm its compatibility with operational environments. Further results are provided in Appendix H and I.

## 4.1. Experimental Setup

**Benchmarks.** We evaluate FLAIR on two established real-world benchmarks: **STATS** (STA, 2015) and **JOB-light** (Leis et al., 2018; 2015). STATS contains over 1 million records, while JOB-light, derived from the IMDB dataset, includes 62 million records. We simulate real-world database conditions in our experiments by incorporating varied SQL operations and design scenarios that mirror different levels of concept drift, ranging from mild to severe. Further details are elaborated in Appendix G.1 and G.6.

**Downstream Tasks.** We primarily assess FLAIR's core performance through cardinality estimation (CE) tasks, alongside exploring its capabilities in user-oriented activities like approximate query processing (AQP) and in-database data analytics involving data classification and regression. Further details are available in Appendix G.3.

**Baselines.** We compare FLAIR with predominant families of CE technologies, including the estimator from PostgreSQL (pos, 1996), and SOTA learned approaches for dynamic environments, such as DeepDB (Hilprecht et al., 2019), ALECE (Li et al., 2023b), and DDUp (Kurmanji

---

[1]For simplicity, we solely consider insertion and deletion since the update operation can be decomposed into these operations.

[2]The code and data of FLAIR are available at https://anonymous.4open.science/r/FLAIR-D4DA/

Table 1: Overall performance of cardinality estimation task under concept drift. The best performances are highlighted in bold and underlined, and the second-best are bold only.

| Data | Method | Mild Drift | | | | | | | | | Severe Drift | | | | | | | | |
| | | GMQ | Q-error | | | | P-error | | | | GMQ | Q-error | | | | P-error | | | |
| | | | 50% | 75% | 90% | 95% | 50% | 75% | 90% | 95% | | 50% | 75% | 90% | 95% | 50% | 75% | 90% | 95% |
| STATS | Fine-tune† | **5.35** | 3.47 | 9.32 | 33.99 | 77.93 | 6.21 | 17.72 | 54.21 | 111.85 | **5.02** | 3.35 | 6.44 | 17.44 | 65.76 | 10.24 | 55.92 | 255.82 | 927.08 |
| | PostgreSQL | 174.38 | 497.56 | 611.53 | 21556.35 | 70977.46 | 8.87 | 52.29 | 157.93 | 174.24 | 293.47 | 758.89 | 6740.46 | 62020.12 | 218196.66 | 10.39 | 83.37 | 401.75 | 1296.15 |
| | ALECE | 20.29 | 15.03 | 52.26 | 197.61 | 430.69 | 7.67 | 30.05 | 131.25 | 249.24 | 36.16 | 22.77 | 112.79 | 624.31 | 1172.69 | 8.61 | 48.72 | 312.12 | 1130.76 |
| | DDUp | 5.79 | 4.49 | 10.20 | 26.41 | 72.68 | 8.00 | 29.59 | 64.91 | 241.27 | 10.95 | 9.51 | 20.22 | 46.15 | 87.18 | 13.61 | 43.92 | 109.24 | 216.64 |
| | FLAIR | **4.49** | 2.86 | 6.93 | 24.06 | 60.94 | 7.01 | 28.04 | 61.70 | 162.61 | **5.47** | 3.12 | 7.87 | 28.52 | 81.57 | 7.97 | 26.78 | 308.43 | 1005.64 |
| Job-light | Fine-tune† | **2.45** | 1.36 | 2.05 | 9.28 | 20.47 | 1.09 | 2.56 | 3.42 | 4.09 | **8.09** | 2.31 | 9.68 | 57.51 | 5168.29 | 1.02 | 1.08 | 1.74 | 1.86 |
| | PostgreSQL | 9.36 | 1.89 | 6.93 | 21.42 | 87.12 | 1.28 | 2.14 | 3.98 | 7.06 | 32.09 | 11.75 | 282.67 | 3834.32 | 7200.49 | 1.90 | 2.78 | 4.20 | 62.43 |
| | DeepDB | 32.28 | 10.52 | 436.77 | 698.12 | 6894.09 | 1.98 | 18.19 | 36.24 | 126.55 | 49.69 | 14.76 | 972.51 | 7864.98 | $7.65e^5$ | 1.77 | 17.31 | 22.31 | 51.89 |
| | ALECE | 12.21 | 11.59 | 19.34 | 26.40 | 63.37 | 1.96 | 4.83 | 8.72 | 19.06 | 27.32 | 11.72 | 114.32 | 1920.34 | 6970.01 | 1.56 | 2.35 | 3.68 | 4.26 |
| | DDUp | 4.16 | 3.60 | 4.99 | 15.62 | 46.98 | 1.59 | 2.14 | 3.88 | 5.12 | 10.96 | 6.65 | 35.35 | 162.51 | 203.34 | 1.09 | 1.65 | 1.89 | 2.79 |
| | FLAIR | **2.36** | 1.29 | 2.09 | 6.93 | 18.62 | 1.18 | 1.36 | 2.94 | 3.67 | **7.95** | 2.38 | 10.21 | 73.91 | 4826.64 | 1.03 | 1.41 | 1.78 | 2.38 |

† Fine-tune typically represents the performance upper bound among baselines, achieved through costly model updates via parameter retraining.

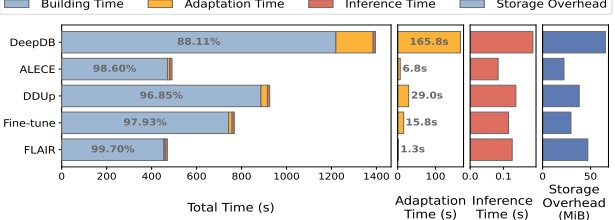

Figure 4: Comparison of model efficiency.

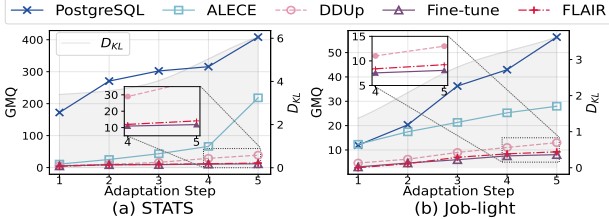

Figure 5: Comparison of model robustness for long-term incremental concept drift.

& Triantafillou, 2023) with NeuroCard (Yang et al., 2020) being used as its base model. We also compare FLAIR with model fine-tuning outlined in (Kurmanji & Triantafillou, 2023), serving as a high-performance baseline despite being computationally intensive. For AQP, our baselines include DBest++ (Ma et al., 2021), which utilizes only frequency tables (FTs) for the update, DBest++FT, which updates both FTs and mixture density networks (MDNs), and DDUp, which uses DBest++ as its base model. For in-database data analytics, we compare FLAIR with AutoML system AutoGluon (Erickson et al., 2020) and established ML algorithms, including K-nearest-neighbors (KNN), RandomForest, MLP, and popular boosting methods, XGBoost (Chen & Guestrin, 2016), LightGBM (Ke et al., 2017) and CatBoost (Prokhorenkova et al., 2018) for data classification, and AutoGluon, SVR, MLP, DecisionTree, RandomForest, and GradientBoosting for regression. See Appendix G.2 and G.4 for baselines and implementation details.

**Evaluation Metrics.** We evaluate FLAIR's effectiveness and efficiency across various tasks using targeted metrics. (1) Effectiveness Metrics: For CE tasks, we report the accuracy by the geometric mean of the Q-error (GMQ) as (Li et al., 2022; Dutt et al., 2019) along with Q-error and P-error across various quantiles, with particular emphasis on the tail performance. For AQP tasks, we use mean relative error (MRE) to evaluate the accuracy of query approximations. Additionally, we apply accuracy and F1 score for data classification and mean squared error (MSE) and the coefficient of determination ($R^2$) for data regression. (2) Efficiency Metrics: We assess FLAIR's efficiency by examining storage overhead, building time, inference time, and adaptation time. See Appendix G.5 for more details on the metrics.

## 4.2. Effectiveness

In Table 1, we report the overall performance comparison in CE task. The results reveal that FLAIR consistently delivers superior performance across all datasets and dynamic scenarios, often matching or even surpassing the outcomes of the fine-tune approach. Specifically, FLAIR achieves the best performance in 29 out of 32 quantile metrics. Even when including fine-tune comparisons, FLAIR surpasses nearly half of the evaluations for all metrics, underscoring its considerable precision in dynamic environments. Additionally, FLAIR significantly outperforms PostgreSQL across all datasets and settings, highlighting the limitations of PostgreSQL's independence assumption that often results in inaccuracies with non-uniform data distributions. Furthermore, our experiments reveal that existing methods, including those using fine-tuning and knowledge distillation, struggle with rapid and complex changes in dynamic systems. In contrast, FLAIR excels by promptly adapting to current concepts during concept drift, without data recollection, offline updates, or separate drift detection processes.

## 4.3. Efficiency

We evaluate the construction efficiency and resource usage of FLAIR alongside baseline models on the JOB-light benchmark. The results in Figure 4 demonstrate that FLAIR is notably efficient in both building and adaptation phases. Remarkably, FLAIR accelerates adaptation speed by $5.2\times$ while reducing the GMQ by 22.5% compared with the best baseline. To further improve FLAIR's inference efficiency, we implement an embedding caching mechanism in FLAIR,

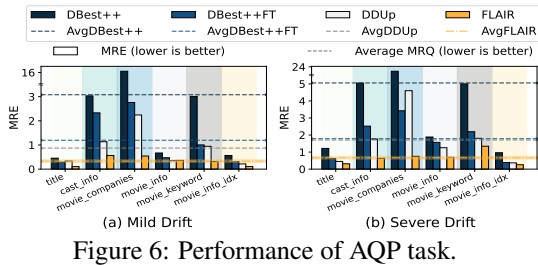

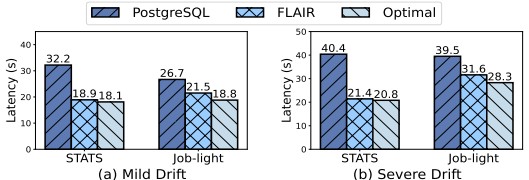

Figure 6: Performance of AQP task.

Figure 8: Comparison of query latency.

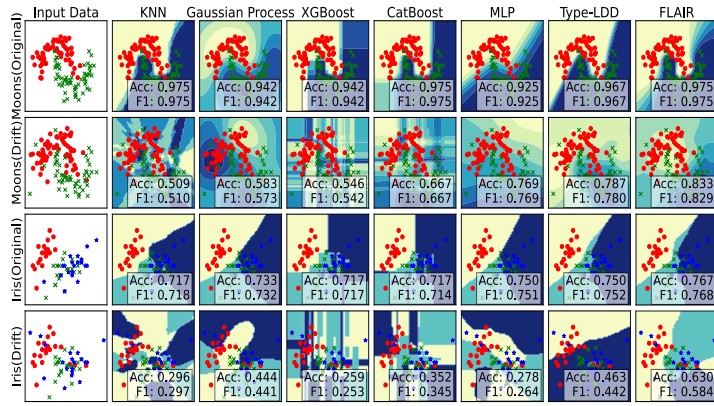

Figure 7: Decision boundaries and model performance on data classification task under concept drift.

which eliminates redundant computations by preventing recomputation on the repeated inputs. This enhancement significantly accelerates the inference process, yielding competitive inference times. Taking the overall performance into consideration, the slightly higher storage requirement imposed by FLAIR is acceptable.

## 4.4. Long-term Incremental Concept Drift

To further assess FLAIR's adaptability, we track the performance on STATS and JOB-light, focusing on gradual drift indicated by rising Kullback-Leibler divergence $D_{KL}$ over extended periods. Figure 5 illustrates that FLAIR effectively handles the challenging conditions of long-term incremental concept drift across both benchmarks, even on par with model fine-tuning. Furthermore, we observe that DDUp based on knowledge distillation is inferior to fine-tuning under long-term gradual drift. This is in line with the results in Section 4.2, highlighting the inherent limitations of knowledge distillation: it mitigates catastrophic forgetting by preserving prior learned knowledge but can inadvertently replicate past errors, whereas fine-tuning directly adjusts to new data, correcting inaccuracies and adapting to evolving distributions. Conversely, FLAIR's innovative in-context adaptation paradigm, guided by dynamic context memory, achieves negligible error accumulation and ensures sustained adaptability without further training, distinguishing it from both knowledge distillation and fine-tuning.

## 4.5. Transferability

In data systems, system-internal tasks like CE provide immediate critical outcomes for optimization, while it is often not straightforward for user-oriented tasks. Next, we validate FLAIR's performance in user-oriented scenarios to showcase its wide applicability, where our context memory establishes a virtuous cycle of user feedback to refine model performance and facilitate system customization.

**Approximate Query Processing.** The results in Figure 6, measured in MRE, consistently show that FLAIR outperforms baseline approaches. Across various relations and dynamic settings, FLAIR achieves significant error reductions, with averages up to or exceeding 10× with DBest++, 3× with DBest++FT, and 2× with DDUp. These findings highlight the effectiveness of FLAIR in handling complex query scenarios. Most of the time, FLAIR outperforms methods that rely on fine-tuning and knowledge distillation, such as DBest++FT and DDUp. This superiority stems from the limitations associated with only updating models during significant data drifts, which may not suffice for the accurate execution of AQP tasks in real and live system scenarios.

**In-database Data Analytics.** We initially conduct a qualitative evaluation on illustrative toy problems to understand the behavior of FLAIR under concept drift, comparing against standard classifiers as shown in Figure 7. We utilize moons and iris datasets from scikit-learn (Pedregosa et al., 2011). For the drift scenarios, we allocate 10% of data for model updates and the remaining 90% for evaluation. In each case, FLAIR effectively captures the decision boundary between samples, delivering well-calibrated predictions. We extend our empirical analysis to real-world tasks, applying data classification for sentiment analysis and data regression for rating prediction on IMDB (See Appendix H).

## 4.6. FLAIR in Action

Given the observation from existing research (Negi et al., 2021; Marcus et al., 2021; Li et al., 2023b) that a smaller Q-error does not necessarily reduce execution times, we extend our investigation by integrating FLAIR into PostgreSQL to assess its efficacy in a full-fledged database system. We evaluate the latency measured as execution time per query on the test set of STATS and JOB-light. As in a recent work (Li et al., 2023b), we substitute PostgreSQL's default cardinality estimator with FLAIR. Specifically, PostgreSQL uses the cardinality estimated by FLAIR to generate the ex-

ecution plan for each query in the benchmarks. The optimal baseline is established by replacing PostgreSQL's built-in estimations with ground-truth cardinalities. As depicted in Figure 8, FLAIR achieves latency that approaches the optimal level based on ground-truth cardinality. Compared to PostgreSQL's built-in cardinality estimator, FLAIR accelerates query execution by up to $1.9\times$. This superiority is even more significant in severe drift scenarios.

## 5. Related Work

**Advances and Challenges of AI×DB.** Database systems are increasingly embracing artificial intelligence (AI), spurring the development of AI-powered databases (AI×DB) (Ooi et al., 2024; Zhu et al., 2024b; Zhao et al., 2025). This fusion marks a new era for database systems, in which AI functionalities are incorporated to enhance the overall system performance and usability. Consequently, advanced models such as deep neural networks (DNNs) and large language models (LLMs) are increasingly being integrated into database systems and applications, which have improved database management such as database tuning (Lao et al., 2024; Huang et al., 2024; Trummer, 2022), cardinality estimation (Lee et al., 2024; Kurmanji & Triantafillou, 2023; Li et al., 2023b; Hilprecht et al., 2019), and indexing (Zhang et al., 2024a; Li et al., 2020; 2023a; Gao et al., 2023; Sun et al., 2023; Zhang et al., 2024b). Recent work (Zeighami & Shahabi, 2024) presents a theoretical foundation for developing machine learning approaches in database systems. However, unlike the data that AI models have been designed for, online transactional processing (OLTP) data is dynamic in nature and such dynamicity affects the robustness of models. Indeed, the phenomenon of *concept drift*, where the underlying data distributions and relations shift, remains a critical challenge. In this study, our goal is to provide a solution for addressing *concept drift* in databases, ensuring both accuracy and sustainability in dynamic environments.

**Model Adaptation in Concept Drift.** Variations in data critically affect the efficacy of AI-powered database systems, also known as learned database systems. Such discrepancies between training data and those encountered post-deployment significantly degrade system performance, challenging model reliability in dynamical environments for the practical deployment (Negi et al., 2023; Zeighami & Shahabi, 2024). Recent cutting-edge machine learning paradigms such as transfer learning (Jain et al., 2023; Kurmanji & Triantafillou, 2023; Zhu et al., 2023; Kurmanji et al., 2024; Ying et al., 2018), active learning (Ma et al., 2020; Li et al., 2022; Lampinen et al., 2024; Zhang et al., 2022), and multi-task learning (Kollias et al., 2024; Wu et al., 2021; Hu et al., 2024) have been employed to mitigate challenges of concept drift in learned database systems. Notably, Kurmanji et al. utilize knowledge distillation, guided by loss-based out-of-distribution data detection for handling data insertions (Kurmanji & Triantafillou, 2023), and explore transfer learning for machine unlearning to address data deletions (Kurmanji et al., 2024). Additionally, reinforcement learning (RL) has been used to strategically reduce the high costs of data collection by allowing an RL agent to selectively determine which subsequent queries to execute in a more targeted fashion (Zhang et al., 2019; Hilprecht et al., 2020; Zheng et al., 2024; Wang et al., 2023b). These strategies, while aimed at improving generalization in fluctuating settings, inherently face critical issues due to their requirements for data recollection and model retraining. For instance, optimizing query performance necessitates executing numerous query plans, a process that is computationally intensive and significantly extends execution time (Wu et al., 2021; Hilprecht & Binnig, 2021; Li et al., 2022). The need for repetitive executions, whenever new concepts are detected, further compounds the operational challenges.

Inspired by large language models (LLMs), zero-shot learning has been employed to enhance model adaptability to dynamic environments and generalize across different tasks (Zhou et al., 2023; Zhu et al., 2024a; Urban et al., 2023; Zhang et al., 2024c; Lin et al., 2025; Li et al., 2025). While this approach is theoretically promising, it faces practical challenges, as pre-training or fine-tuning foundation models still requires substantial real-world data collection. Additionally, the quality and relevance of training data to actual workloads remain uncertain until deployment, making post-deployment performance unpredictable. Further, existing methods struggle to keep pace with real-time evolving concepts and overlook inter-query relations, which compromises their effectiveness. To fundamentally address these challenges, we propose a fresh perspective on online adaptation for database systems that supports on-the-fly in-context adaptation to evolving concepts without unnecessary data collection or retraining, ensuring unparalleled effectiveness and efficiency in operational settings.

## 6. Conclusions

With frequent updates, the data in the database evolves, resulting in concept drift. Learned database operations are susceptible to concept drift, and may suffer significant prediction accuracy losses. This paper presents a novel online adaptation framework called FLAIR, which can adapt the in-database predictive model to evolving concepts automatically without cumbersome data recollection and model retraining. FLAIR performs Bayesian meta-training using abundant synthetic data sampled from dynamic task distributions. After meta-training, it generates adapted predictions by prompting the model with contextual information regarding the current concept. Extensive experiments across various database operations demonstrate that FLAIR is effective, efficient, and transferable in dynamic data systems.

## ACKNOWLEDGMENTS

The work of BIT researchers is partially supported by National Science and Technology Major Project under Grant 2022ZD0119701, National Natural Science Foundation of China National Science Fund for Distinguished Young Scholars under Grant 62025301, and National Natural Science Foundation of China National Science Fund for Young Scientists (Ph.D.) under Grant 624B2027. The work of NUS researchers is supported by the Lee Foundation in terms of Beng Chin Ooi's Lee Kong Chian Centennial Professorship fund. Yanyan Shen's work is supported by National Key Research and Development Program of China under Grant 2022YFE0200500. Gang Chen's work is supported by National Key Research and Development Program of China under Grant 2022YFB2703100.

## Impact Statement

This paper showcases advances in the integration of Machine Learning within data systems, presenting a framework that enhances the reliability and usability of Machine Learning in real-world dynamic environments. By facilitating various database operations, the proposed framework encourages broader adoption in practice, potentially reducing operational costs and energy consumption. There can be various societal consequences, including enhancing decision-making in data-reliant sectors such as healthcare, finance, and public services, and enabling businesses to achieve significant cost reductions through optimized operations and minimized manual intervention.

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

# APPENDIX

## A. Notation Table

In this paper, scalars are denoted by symbols such as $x$, vectors are represented by boldface symbols such as $\mathbf{x}$, and matrices are described by uppercase boldface symbols such as $\mathbf{X}$. To provide a comprehensive overview of the notations used throughout the paper, we present a summary of notations in Table 2 as a quick reference to facilitate the understanding and recall of each symbol.

Table 2: Notations.

| Notation | Description |
|---|---|
| $\mathbf{R_i}$ | The $i$th relation in the database. |
| $\mathbf{a_n^i}$ | The $n$th attribute of the relation $\mathbf{R_i}$. |
| $\mathbf{x_n^i}$ | The histogram for attribute $\mathbf{a_n^i}$. |
| $n_i$ | Number of attributes in relation $\mathbf{R_i}$. |
| $\mathbf{X}$ | Data matrix, aggregated by attribute histograms $\mathbf{x_n^i}$ across all relations in the database. |
| $\delta$ | Number of histogram bins, a hyperparameter that controls the granularity and dimensionality of $\mathbf{X}$. |
| $\mathbf{q}_{\mathcal{Q}} = <\mathbf{q}_J, \mathbf{q}_F>$ | Query vector, formed by concatenating join predicate vector $\mathbf{q}_J$ and filter predicate vector $\mathbf{q}_F$. |
| $\mathcal{D}_t$ | Data distribution at time $t$. |
| $f_{\mathcal{D}_t}$ | Mapping function of data distribution $\mathcal{D}_t$. |
| $d_k$ | Scaling factor for the dot product in the attention mechanism. |
| $M$ | Number of attention heads. |
| $\mathbf{q}^m, \mathbf{k}^m, \mathbf{v}^m$ | The query, key, value vector for the $m$th attention head. |
| $\mathbf{W}_q^m, \mathbf{W}_k^m, \mathbf{W}_v^m$ | Transformation matrices for query, key, and value in the $m$th attention head. |
| $\mathbf{W}_o$ | Output transformation matrix. |
| $\mathbf{Z}_{\mathcal{O}}$ | Final output of the MHSA phase. |
| $\mathbf{z}_{\mathcal{T}}$ | Task vector obtained by the TFM. |
| $d_a$ | The attention dimension in the DDE. |
| $\mathcal{Q}_{pmt}$ | The queue contains $\varrho$ recent queries processed by the system. |
| $\mathcal{Y}_{pmt}$ | The queue contains the corresponding system outputs of $\mathcal{Q}_{pmt}$. |
| $\mathcal{C} = (\mathcal{Q}_{pmt}, \mathcal{Y}_{pmt})$ | Context memory, containing contextual information of the current task. |
| $\varrho$ | Size of the context memory. |
| $\Phi$ | Prior distributions in Bayesian meta-training. |
| $\mathcal{L}_{DDE}$ | Negative log likelihood loss of the DDE. |
| $\mathcal{L}_{reg}$ | The instantiation of $\mathcal{L}_{DDE}$ for regression tasks. |
| $\mathcal{L}_{cls}$ | The instantiation of $\mathcal{L}_{DDE}$ for classification tasks. |
| $\mathcal{L}_{TS}$ | Task-specific loss of the TFM. |
| $\eta_{\mathcal{T}}, \eta_{\mathcal{D}}$ | Learning rates for the TFM and DDE modules. |
| $\Theta_{\mathcal{T}}, \Theta_{\mathcal{D}}$ | Model parameters for the TFM and DDE modules. |

# B. Extended Preliminaries

In this section, we further explore the technical aspects of two techniques central to FLAIR, in-context learning and Prior-data Fitted Networks.

**In-context Learning.** Emerging as a revolutionary paradigm alongside the advancement of large language models (LLMs), *in-context learning* (ICL) enables the foundation models to make direct predictions based on contextual examples without the need for parameter updates (Baldassini et al., 2024; Radford et al., 2019; Raffel et al., 2020; Brown et al., 2020; Achiam et al., 2023; Jeon et al., 2024), akin to human reasoning processes. Specifically, ICL empowers a model by conditioning it on a few select set of input-output examples. This setup estimates the likelihood of potential outputs by utilizing contextual demonstrations with a sophisticated pre-trained model Formally, given an input $\mathbf{x}$ and a set of candidate outputs $Y = \{y_1, y_2, \cdots\}$, a pre-trained model $\mathcal{M}$ selects the candidate output with the highest score as its prediction, conditioned on a demonstration set $\mathcal{C}$. The demonstration set $\mathcal{C}$ includes $k$ examples, represented as $\mathcal{C} = \{(\mathbf{x}_1, y_1), \cdots, (\mathbf{x}_k, y_k)\}$, where $(\mathbf{x}_k, y_k)$ is an example of input-output pair pertinent to the task. The likelihood of a candidate output $y_i$ is determined by the scoring function $\mathcal{S}$ applied to the entire input examples using the model $\mathcal{M}$. The predicted output $\hat{y}$ is then the candidate with the highest probability:

$$P(y_i|\mathbf{x}) \triangleq \mathcal{S}_{\mathcal{M}}(y_i, \mathbf{x}, \mathcal{C}) \tag{17}$$

$$\hat{y} = \arg\max_{y_i \in Y} P(y_i|\mathbf{x}) \tag{18}$$

The scoring function $\mathcal{S}$ evaluates the plausibility of each candidate output based on the demonstration set $\mathcal{C}$ and the query input $\mathbf{x}$, supporting various learning scenarios or new tasks without specific training.

In FLAIR, we leverage the principles of ICL to enable context-aware adaptation for dynamic data systems. Our approach allows the model to dynamically adjust its predictions based on evolving concepts rather than relying on a static input-output mapping fixed to a particular data distribution, delivering superior modeling performance without the need for periodic model retraining or fine-tuning, as required by existing methods.

**Prior-data Fitted Networks.** *Prior-data Fitted Networks* (PFNs) (Müller et al., 2022; Hollmann et al., 2023; Helli et al., 2024) are advanced classifiers for tabular data, designed to perform Bayesian inference by pre-training on synthetic datasets, which are constructed based on a carefully designed prior distribution. This approach allows PFNs to make accurate predictions on new, unseen data without the need for further parameter updates, thereby effectively approximating the Posterior Predictive Distribution (PPD). In Bayesian learning, the PPD estimates the likelihood of predictions for new data points based on observed data $D$ and a prior distribution of hypotheses $\Phi$. For a given test sample $\mathbf{x_t}$, PFNs calculate the distribution of the label $y_t$ as follows:

$$p(y_t|\mathbf{x_t}, D) \propto \int_{\Phi} p(y|\mathbf{x_t}, D)p(D|\phi)p(\phi)d\phi \tag{19}$$

where $\phi \in \Phi$ represents a specific hypothesis, and $p(D|\phi)$ is the likelihood of observing the data $D$ given the hypothesis $\phi$. The PPD integrates over all hypotheses, weighted by respective priors and data likelihoods, thus enabling PFNs to make informed probabilistic predictions. By approximating PPD, PFNs merge Bayesian inference with deep learning to enhance the accuracy of predictions for diverse applications. In FLAIR, we integrate context into the input, via a task featurization module, and then, model the current data distribution $p(D|\phi)$ explicitly by uncovering PFNs to adaptive cases through a dynamic decision engine, enhancing prediction accuracy for $p(y_t|\mathbf{x_t}, D)$. This enables FLAIR to adapt to new concepts in databases on-the-fly.

# C. Data and Task Query Encoding

To encode the data, the process first obtains a unified representation of the data distribution within the database via histogram encoding of each attribute (i.e., column) across all relations (i.e., tables). Specifically, each attribute $\mathbf{a_n^i}$ within a relation $\mathbf{R_i}$ is represented by a histogram defined as $\mathbf{x}_n^i = [x_1, \cdots, x_\delta]$, where $\delta$ indicates the number of bins, a parameter that can be adjusted to account for the complexity of the data distribution. These histograms are aggregated into a set $\{\mathbf{x_n^i} | 1 \leqslant n \leqslant n_i, 1 \leqslant i \leqslant N, n, i \in \mathbb{Z}\}$ after scaling into $[0, 1]$, where $n_i$ is the number of attributes in relation $\mathbf{R_i}$, and $N$ is the total number of relations. The set is then aggregated into *data vectors* $X_D$ of dimension $\delta \times \sum_{i=1}^{N} n_i$, offering a holistic view of the entire database.

Subsequently, we encode the task query formulated as:

```
SELECT AGG FROM R₁,...,Rₙ
WHERE join predicates ⋈ AND filter predicates π;
```

Here, `AGG` represents the aggregate function such as `COUNT`, `SUM`, or `AVG`. Join predicates, formatted as $\mathbf{R_i a_{n_i}^i} = \mathbf{R_j a_{n_j}^j}$ are converted into binary vectors $\mathbf{q}_J$ by a one-hot encoding-like strategy. For filter predicates formatted as $\mathbf{R_i a_{n_i}^i}$ op $\mho$, where op $\in \{<, \leqslant, \geqslant, >, =\}$ denotes comparison operators and $\mho$ is the condition value. We encode them into $\mathbf{q}_F$ by converting conditions on attributes into two boundary values, forming a $2\sum_{i=1}^N n_i$ dimensional vector. The final *query vector* $\mathbf{q}_Q = <\mathbf{q}_J, \mathbf{q}_F>$ is obtained by concatenating the join vector $\mathbf{q}_J$ and filter vector $\mathbf{q}_F$, capturing pertinent structural and conditional information of the task query.

## D. FLAIR Workflow: Training to Inference

As outlined in Section 2, FLAIR $\mathcal{M}_F$ comprises two cascading modules, the TFM module $\mathcal{M}_{TFM}$ parameterized by $\Theta_{\mathcal{T}}$ and the DDE module $\mathcal{M}_{DDE}$ parameterized by $\Theta_{\mathcal{D}}$, represented as follows:

$$\mathcal{M}_F(\mathbf{x}; \Theta_{\mathcal{T}}, \Theta_{\mathcal{D}}) = \mathcal{M}_{DDE}(\mathcal{M}_{TFM}(\mathbf{x}; \Theta_{\mathcal{T}}); \Theta_{\mathcal{D}}) \tag{20}$$

Next, we elaborate on the workflow of FLAIR, covering phases of offline training and online inference and adaptation.

**Offline Training.** FLAIR is trained in two stages, as outlined in Algorithm 1. (i) In the first stage, the DDE module $\mathcal{M}_{DDE}$ undergoes a *one-off meta-training* phase using the loss function $\mathcal{L}_{DDE}$ formalized in Eq. 10 across various task distributions. These distributions are generated based on crafted priors, tailored to encompass a broad spectrum of scenarios rather than specific real-world data, which enables DDE to generalize across various tasks. Note that the Bayesian meta-training is not to optimize FLAIR directly on end tasks but to prepare DDE to adapt to new tasks met during inference without further training. In all our experiments across diverse tasks, we perform Bayesian meta-training only once, demonstrating its efficiency and scalability in real-world applications. (ii) In the second stage, the $\mathcal{M}_{TFM}$ module is trained to extract informative latent features that are critical for the specific tasks at hand. The training of TFM is tailored to optimize performance on these tasks, utilizing a task-specific loss function $\mathcal{L}_{TS}$ to extract standardized and informative features for the DDE module. In particular, for tasks outputting raw logits, such as cardinality estimation and approximate query processing, a mean squared error (MSE) loss will be employed to optimize the representation space. Otherwise, we utilize cross-entropy loss for tasks generating probability distributions via softmax activation, like in-database data classification. Overall, the meta-training phase is not confined to specific tasks, instead, it establishes a foundation for efficient adaptation by learning a flexible and generalizable parameter space. Meanwhile, the TFM is independently tuned for the specific task at hand. Together, this offline training approach ensures that the two key modules of FLAIR work seamlessly to the adaptability of the model in dynamic databases.

---

**Algorithm 1** FLAIR Training

---

**Input:** Designed priors $p(\phi)$, number of synthetic datasets $\mathcal{H}$, each with $N_o$ observed samples, queue size $\varrho$ in the context memory, learning rate $\eta_{\mathcal{T}}$ for $\mathcal{M}_{TFM}$ and $\eta_{\mathcal{D}}$ for $\mathcal{M}_{DDE}$.
**Output:** FLAIR $\mathcal{M}_F(\mathbf{x}; \Theta_{\mathcal{T}}, \Theta_{\mathcal{D}})$ constructed by cascading $\mathcal{M}_{TFM}$ and $\mathcal{M}_{DDE}$ with parameters $\Theta_{\mathcal{T}}$ and $\Theta_{\mathcal{D}}$.
 1: Initialize $\mathcal{M}_{TFM}$ and $\mathcal{M}_{DDE}$ with random weights $\Theta_{\mathcal{T}}$ and $\Theta_{\mathcal{D}}$
 2: **for** $i = 1$ **to** $\mathcal{H}$ **do**
 3:     Sample synthetic datasets $\widetilde{D}_i \sim p(\mathcal{C}|\phi)$
 4:     Randomly select context $\mathcal{C}$ based on $\{(\mathbf{x}_j, \mathbf{y}_j)\}_{j=1}^{\varrho}$ from $\widetilde{D}_i$
 5:     **repeat**
 6:         Randomly select a training batch $\{(\mathbf{x}_j, \mathbf{y}_j)\}_{j=1}^{N_o}$ from $\widetilde{D}_i$
 7:         Compute stochastic loss $\mathcal{L}_{DDE}$ using Eq. 10
 8:         Update $\Theta_{\mathcal{D}}$ using stochastic gradient descent $\Theta_{\mathcal{D}} \leftarrow \Theta_{\mathcal{D}} - \eta_{\mathcal{D}} \nabla_{\Theta_{\mathcal{D}}} \mathcal{L}_{DDE}$
 9:     **until** Convergence
10: **end for**
11: **repeat**
12:     Randomly sample a minibatch
13:     Update $\Theta_{\mathcal{T}}$ by minimizing the loss $\mathcal{L}_{TS}$ of the specific task $\Theta_{\mathcal{T}} \leftarrow \Theta_{\mathcal{T}} - \eta_{\mathcal{T}} \nabla_{\Theta_{\mathcal{T}}} \mathcal{L}_{TS}$
14: **until** Convergence
15: $\mathcal{M}_F(\mathbf{x}; \Theta_{\mathcal{T}}, \Theta_{\mathcal{D}}) = \mathcal{M}_{DDE}(\mathcal{M}_{TFM}(\mathbf{x}; \Theta_{\mathcal{T}}); \Theta_{\mathcal{D}})$;
16: **Return** FLAIR $\mathcal{M}_F$

---

**Online Inference and Adaptation.** Once trained, FLAIR is ready for deployment in a real-time environment, where it performs concurrent online inference and adaptation under evolving concepts as detailed in Algorithm 2. Specifically, for an incoming input query $\mathbf{x}$, the TFM first extracts its task vector as shown in Eq. 21 below. The task vector, along with its contextual information in context memory $\mathcal{C} = (\mathcal{Q}_{pmt}, \mathcal{Y}_{pmt})$, are then fed to the DDE module as Eq. 22. After executing the current query, the query and the corresponding ground-truth result returned by the database are used to update the queues in context memory, i.e., $\mathcal{Q}_{pmt}$ and $\mathcal{Y}_{pmt}$, respectively. Thus, FLAIR effectively utilizes contextual information from the context memory to adapt to tasks encountered during inference.

$$\mathbf{z}_{\mathcal{T}} = \mathcal{M}_{TFM}(\mathbf{x}; \Theta_{\mathcal{T}}) \tag{21}$$

$$\mathbf{y} = \mathcal{M}_{DDE}(\mathbf{z}_{\mathcal{T}}, \mathcal{C}; \Theta_{\mathcal{D}}) \tag{22}$$

---

**Algorithm 2** Concurrent FLAIR Inference and Adaptation

---

**Input:** $\mathcal{M}_{TFM}$ and $\mathcal{M}_{DDE}$ with parameters $\Theta_{\mathcal{T}}$ and $\Theta_{\mathcal{D}}$, input query and data underlying the data system.
**Output:** Predicted output $\mathbf{y}$.
1: Extract latent feature $\mathbf{z}_{\mathcal{T}}$ incorporating information from query and data, using $\mathcal{M}_{TFM}$ as $\mathbf{z}_{\mathcal{T}} = \mathcal{M}_{TFM}(\mathbf{x}; \Theta_{\mathcal{T}})$
2: Gather context memory $\mathcal{C} = (\mathcal{Q}_{pmt}, \mathcal{Y}_{pmt})$
3: Predict $\mathbf{y}$ by inputting latent feature $\mathbf{z}_{\mathcal{T}}$ and context memory $\mathcal{C}$ into $\mathcal{M}_{DDE}$ as $\mathbf{y} = \mathcal{M}_{DDE}(\mathbf{z}_{\mathcal{T}}, \mathcal{C}; \Theta_{\mathcal{D}})$
4: Store $\mathbf{z}_{\mathcal{T}}$ and the corresponding system output $\mathbf{y}^*$ into queue $\mathcal{Q}_{pmt}$ and $\mathcal{Y}_{pmt}$ to update the context memory $\mathcal{C}$
5: Remove oldest entries from $\mathcal{Q}_{pmt}, \mathcal{Y}_{pmt}$ to maintain size $\varrho$
6: **Return** $\mathbf{y}$

---

# E. Proofs

## E.1. Proof of Theorem 3.1

*Proof.* Let $Y_{ins}$ be the number of inserted data points out of $k$ operations that land in query $\mathbf{x}$ and $Y_{del}$ be the number of deleted data points out of $k$ operations that come from $\mathbf{x}$, where $\mathbf{x}$ specifies an axis-parallel hyperrectangle within the data space. Then, the final frequency of points in $\mathbf{x}$ is

$$f_{D^j}(\mathbf{x}) = \sum_{d \in D^j} \mathbf{I}_{d \in \mathbf{x}} = f_{D^i}(\mathbf{x}) + Y_{ins} - Y_{del}$$

Set $\mathcal{Y} = Y_{ins} - Y_{del}$, then each data operation can increment $\mathcal{Y}$ by $+1$ (insert), decrement $\mathcal{Y}$ by $-1$ (delete), or leave $\mathcal{Y}$ unchanged if the inserted or deleted data point is out of $\mathbf{x}$. We can represent the net difference for each single-point data operation as $\mathcal{Y}(\omega) = \sum_{t=1}^{k} Y_t(\omega)$, where $Y_t(\omega)$ takes values in $\{-1, 0, 1\}$. Consequently, $\mathcal{Y}$ is an integer-valued random variable in $[-k, k]$. Hence, the generalization error can be represented as

$$\mathbb{E}_{D^j \sim \chi}\big[|\hat{f}_i(\mathbf{x}) - f_{D^j}(\mathbf{x})|\big] = \mathbb{E}_{D^j \sim \chi}\big[|\hat{f}_i(\mathbf{x}) - (f_{D^i}(\mathbf{x}) + Y_{ins} - Y_{del})|\big]$$
$$= \mathbb{E}_{D^i \sim \chi}\big[\mathbb{E}_{D^{i:j} \sim \chi}\big[|(\hat{f}_i(\mathbf{x}) - f_{D^i}(\mathbf{x})) - \mathcal{Y}| \,|D^i|\big]\big]$$

Observe that $\hat{f}_i(\mathbf{x}) - f_{D^i}(\mathbf{x})$ is a fixed quantity for a given $D^i$. Recall the classic fact that $\arg\min_c \mathbb{E}[|X - c|] = \text{Med}(X)$ for any random variable $X$ (Wasan, 1970). Thus, for each realized dataset $D^i$, the fixed quantity $\hat{f}_i(\mathbf{x}) - f_{D^i}(\mathbf{x})$ can be seen as an offset and $\mathcal{Y}$ is the random part, so we have

$$\mathbb{E}_{D^{i:j} \sim \chi}\big[|(\hat{f}_i(\mathbf{x}) - f_{D^i}(\mathbf{x})) - \mathcal{Y}|\big] \geqslant \mathbb{E}_{D^{i:j} \sim \chi}\big[|\text{Med}(\mathcal{Y}) - \mathcal{Y}|\big]$$

To assess the worst-case scenario, we consider the extreme outcomes in which $\mathcal{Y}$ attains $k$ or $-k$. Consequently, $\text{Med}(\mathcal{Y})$ is forced to be either 1 or $-1$. In both extremes, $|\text{Med}(\mathcal{Y}) - \mathcal{Y}|$ reveals a mismatch of $k - 1$. Maximizing over all possible queries $\mathbf{x}$ thus enforces

$$\sup_{\mathbf{x}} \mathbb{E}_{D^j \sim \chi}\big[|\hat{f}_i(\mathbf{x}) - f_{D^j}(\mathbf{x})|\big] \geqslant k - 1$$

$\square$

### E.2. Proof of Theorem 3.2

*Proof.* First, consider

$$\mathbb{E}_{D^j \sim \chi}\Big[\big|\hat{f}(\mathbf{x}|\mathcal{C}^j) - f_{D^j}(\mathbf{x})\big|\Big] = \mathbb{E}_{D^j \sim \chi}\Big[\big|\hat{f}(\mathbf{x}|\mathcal{C}^j) - f_{\mathcal{C}^j}(\mathbf{x}) + f_{\mathcal{C}^j}(\mathbf{x}) - f_{D^j}(\mathbf{x})\big|\Big]$$

where $f_{\mathcal{C}^j}(\mathbf{x})$ denotes the optimal model updated by $\mathcal{C}^j$ after training on the initial data $D^i$. By introducing $f_{\mathcal{C}^j}(\mathbf{x})$ term, we decompose the maximum generalization error of FLAIR into two parts using the triangle inequality.

$$\mathbb{E}_{D^j \sim \chi}\Big[\big|\hat{f}(\mathbf{x}|\mathcal{C}^j) - f_{D^j}(\mathbf{x})\big|\Big] \leqslant \mathbb{E}_{D^j \sim \chi}\Big[\big|\hat{f}(\mathbf{x}|\mathcal{C}^j) - f_{\mathcal{C}^j}(\mathbf{x})\big|\Big] + \mathbb{E}_{D^j \sim \chi}\Big[\big|f_{\mathcal{C}^j}(\mathbf{x}) - f_{D^j}(\mathbf{x})\big|\Big]$$

The first term $\mathbb{E}_{D^j \sim \chi}\Big[\big|\hat{f}(\mathbf{x}|\mathcal{C}^j) - f_{\mathcal{C}^j}(\mathbf{x})\big|\Big]$ (denoted as $\mathcal{E}_{\mathcal{M}}$) represents the error between the FLAIR output $\hat{f}(\mathbf{x}|\mathcal{C}^j)$ and the optimal model $f_{\mathcal{C}^j}(\mathbf{x})$ trained on the context memory $\mathcal{C}^j$. This measures the FLAIR's ability to approximate $f_{\mathcal{C}^j}(\mathbf{x})$, reflecting whether the FLAIR can efficiently utilize the information in context memory $\mathcal{C}^j$ for prediction. Theoretically, $\mathcal{E}_{\mathcal{M}}$ tends to be 0 if FLAIR can fully learn the mapping from $\mathcal{C}^j$ to the posterior distribution. Using the generalization error bound on the PAC-Bayes framework (Amit & Meir, 2018), we have

$$\mathcal{E}_{\mathcal{M}} = \mathbb{E}_{D^j \sim \chi}\Big[\big|\hat{f}(\mathbf{x}|\mathcal{C}^j) - f_{\mathcal{C}^j}(\mathbf{x})\big|\Big] \leqslant \sqrt{\frac{\mathrm{KL}(q(f)\|p(f)) + \ln\frac{1}{\delta}}{2\varrho}}$$

with high probability $1 - \delta$. Here, $\mathrm{KL}(q(f)\|p(f))$ is the Kullback-Leibler divergence (Kullback & Leibler, 1951) between the posterior distribution $q(f)$ over the model parameters, conditioned on the context memory $\mathcal{C}^j$, and prior distribution $p(f)$ defined during meta-training. $\delta \in (0, 1)$ is the confidence parameter. This result states that, for each particular $\mathbf{x}$, the expected error of $\hat{f}(\mathbf{x}|\mathcal{C}^j)$ relative to $f_{\mathcal{C}^j}(\mathbf{x})$ is controlled by a PAC-Bayes term of order $\mathcal{O}\big(\frac{1}{\sqrt{\varrho}}\big)$. Assuming sufficient training of FLAIR, $\mathrm{KL}(q(f)\|p(f))$ is bounded by a small constant $\kappa$, leading to

$$\sup_{\mathbf{x}} \mathbb{E}_{D^j \sim \chi}\Big[\big|\hat{f}(\mathbf{x}|\mathcal{C}^j) - f_{\mathcal{C}^j}(\mathbf{x})\big|\Big] \leqslant \sqrt{\frac{\kappa + \ln\frac{1}{\delta}}{2\varrho}}$$

This result highlights that FLAIR's generalization performance improves as the size of the context memory $\varrho$ increases.

Moreover, the second term $\mathbb{E}_{D^j \sim \chi}\Big[\big|f_{\mathcal{C}^j}(\mathbf{x}) - f_{D^j}(\mathbf{x})\big|\Big]$ (denoted as $\mathcal{E}_{\mathcal{C}}$) measures how well the context memory $\mathcal{C}^j$ approximates the full data $D^j$. We assume the performance of $f_{\mathcal{C}^j}(\mathbf{q})$ is estimated by the sample average on $\mathcal{C}^j$, i.e., $\overline{V}_{\mathcal{C}^j} = \frac{1}{\varrho}\sum_{i=1}^{\varrho} V_i$, where $V_i$ is defined as the performance metric of the model on the $i$-th data point in $\mathcal{C}^j$. By Hoeffding's inequality (Hoeffding, 1994), the probability of the deviation between $f_{\mathcal{C}^j}(\mathbf{x})$ and $f_{D^j}(\mathbf{x})$ satisfies

$$\Pr\Big[\big|f_{\mathcal{C}^j}(\mathbf{x}) - f_{D^j}(\mathbf{x})\big| \geqslant \epsilon\Big] = \Pr\Big[\big|\overline{V}_{\mathcal{C}^j} - \mu_{D^j}\big| \geqslant \epsilon\Big] \leqslant 2\exp(-2\varrho\epsilon^2)$$

where $\mu_{D^j}$ denotes the expected performance on $D^j$. Grounded in probability measure theory, the expectation of a non-negative random variable can be computed by integrating the probability that the variable exceeds all possible values (Ross et al., 1976). Consequently, integrating over $\epsilon$, the expected approximation error satisfies

$$\mathcal{E}_{\mathcal{C}} = \mathbb{E}_{D^j \sim \chi}\Big[\big|f_{\mathcal{C}^j}(\mathbf{x}) - f_{D^j}(\mathbf{x})\big|\Big]$$

$$= \int_0^\infty \Pr\Big[\big|f_{\mathcal{C}^j}(\mathbf{x}) - f_{D^j}(\mathbf{x})\big| \geqslant \epsilon\Big] d\epsilon$$

$$\leqslant \int_0^\infty 2\exp(-2\varrho\epsilon^2) d\epsilon$$

$$= 2 \cdot \sqrt{\frac{\pi}{4 \cdot 2\varrho}} = \sqrt{\frac{\pi}{2\varrho}}$$

Finally, adding the two terms $\mathcal{E}_{\mathcal{M}}$ and $\mathcal{E}_{\mathcal{C}}$, we obtain

$$\sup_{\mathbf{x}} \mathbb{E}_{D^j \sim \chi}\Big[\big|\hat{f}(\mathbf{x}|\mathcal{C}^j) - f_{D^j}(\mathbf{x})\big|\Big] \leqslant \mathcal{E}_{\mathcal{M}} + \mathcal{E}_{\mathcal{C}} = \sqrt{\frac{\kappa + \ln\frac{1}{\delta}}{2\varrho}} + \sqrt{\frac{\pi}{2\varrho}} = \Big(\sqrt{\frac{1}{2}\big(\kappa + \ln\frac{1}{\delta}\big)} + \sqrt{\frac{\pi}{2}}\Big)\frac{1}{\sqrt{\varrho}}$$

$\square$

# F. Key Characteristics of FLAIR

We discuss FLAIR on three key aspects that are crucial for deployment, namely effectiveness, efficiency, and transferability.

**Effectiveness.** FLAIR demonstrates robust effectiveness, particularly in handling dynamic environments. The task featurization module of FLAIR prepares standardized task features for the subsequent DDE modeling, providing a holistic perspective to achieve a comprehensive understanding of the context. Furthermore, compared to the existing event-driven adaptation to evolving concepts like gradient-based out-of-distribution (OOD) detection (Kurmanji & Triantafillou, 2023), our FLAIR leverages ongoing access to relevant contextual information for contextualized task modeling, which greatly enhances prediction accuracy under evolving concepts.

**Efficiency.** As opposed to conventional models that require continuous training-based updates to adapt to new concepts, FLAIR supports adaptation on-the-fly, eliminating the need for post-deployment data recollection and model retraining. This capability enables FLAIR to be deployed into operational environments with ease, substantially saving on resources. Specifically, each attribute is initially represented by a histogram of size $\delta$, giving an initialization complexity of $\mathcal{O}(\delta \sum_{i=1}^{N} n_i)$. Subsequent insert, delete, or update operations require only $\mathcal{O}(N_v)$ incremental maintenance on the affected histograms where $N_v$ is the number of the records involved. Set the number of join and filter predicates in the query as $N_J$ and $N_F$. The time complexity of encoding a task query is approximately $\mathcal{O}(N_J + N_F)$, which is typically small and negligible. For simplicity of analysis, we disregard the MLP's computational cost, as it is typically overshadowed by the dominant attention-related operations. Thus, for the task featurization stage, the MHSA yields a complexity of $\mathcal{O}(d_a \, \delta^2)$, where we denote the embedding and attention dimension as $d_a$. The cost reduces to $\mathcal{O}(d_a \, \delta)$ for the MHCA part, because the cross-attention occurs only once between the single query vector and the $\delta$-dimensional data representation. In the DDE, the model employs a FIFO key-value queue of $\varrho$ input-output pairs as a context memory. When a new input vector (viewed as a single token) attends over these $\varrho$ stored entries, the time complexity grows linearly in $\varrho$, i.e., $\mathcal{O}(d_a \, \varrho)$, without incurring additional self-attention among the $\varrho$ memory entries themselves, thus being scalable and avoiding the quadratic overhead.

**Transferability.** Beyond its proven effectiveness and efficiency, the transferability of FLAIR is a pivotal feature for its practical deployment in diverse settings. Central to the design of FLAIR is its in-context adaptation capability, which is inherently task-agnostic and facilitates ready application across various tasks including both regression and classification. This design enables FLAIR to support AI-powered database applications that meet both system-internal functions and user-oriented objectives such as cardinality estimation and approximate query processing. The broad applicability of FLAIR is thoroughly demonstrated in Section 4, providing empirical evidence of its superior utility across diverse tabular data-driven domains.

# G. Details on Experimental Setup

In this section, we detail the experimental setup and provide additional information to facilitate reader comprehension and replication of our study.

## G.1. Benchmarks

We evaluate FLAIR on two real-world benchmarks that are commonly referenced in previous database system studies (Sun & Li, 2019; Yang et al., 2020; Hilprecht et al., 2019; Han et al., 2021; Li et al., 2023b; Zeng et al., 2024).

- **STATS** (STA, 2015), includes 8 relations with 43 attributes. There are 1,029,842 records from the anonymized Stats Stack Exchange network. The benchmark workload includes 146 queries with 2603 sub-queries, featuring both PK-FK and FK-FK join.

- **JOB-light** (Leis et al., 2018; 2015), derives from a subset of the Internet Movie Data Base (IMDB) dataset[3] and encompasses 6 relations with 14 attributes. There are 62,118,470 records in total. The benchmark workload consists of 70 queries with 696 sub-queries, focusing on PK-FK join.

As in a recent work (Li et al., 2023b), our evaluation involves randomly generating 2000 diverse queries with sub-queries to form the training set for each benchmark. In the STATS benchmark, we utilize an existing workload of 146 queries with 2603 sub-queries as the test set. For JOB-light, the test set comprises 70 queries associated with 696 sub-queries.

---

[3]https://www.imdb.com/

Additionally, we incorporate a dynamic workload into each benchmark's training and test sets. This dynamic workload includes various SQL operations, including insert, delete, and update, which are strategically varied in proportion throughout different phases of the experiment. Notably, the ground truth for the queries is obtained by executing them, as both the dynamic workload and data changes can influence the results over time. For the cardinality estimation task, queries yielding a ground-truth cardinality of zero are excluded from the analysis to ensure data integrity and relevance.

### G.2. Baselines

In our experiments, we first compare FLAIR with predominant families of cardinality estimation technologies, including the estimator from PostgreSQL (pos, 1996), and state-of-the-art learned approaches for dynamic environments, such as DeepDB (Hilprecht et al., 2019), ALECE (Li et al., 2023b), and DDUp (Kurmanji & Triantafillou, 2023) with Neuro-Card (Yang et al., 2020) being used as its base model. We also compare FLAIR with model fine-tuning outlined in (Kurmanji & Triantafillou, 2023), which serves as a high-performance baseline despite being computationally intensive. For AQP, our baselines include DBest++ (Ma et al., 2021), which utilizes only frequency tables (FTs) for the update, DBest++FT, which updates both FTs and mixture density networks (MDNs), and DDUp (Kurmanji & Triantafillou, 2023), which uses DBest++ as its base model. For in-database data analytics, we compare FLAIR with AutoML system AutoGluon (Erickson et al., 2020) and established machine learning algorithms, including K-nearest-neighbors (KNN), RandomForest, MLP, and popular tree-based boosting methods, XGBoost (Chen & Guestrin, 2016), LightGBM (Ke et al., 2017) and CatBoost (Prokhorenkova et al., 2018) for data classification, and AutoGluon, SVR, MLP, DecisionTree, RandomForest, and GradientBoosting for regression. We briefly introduce the baselines in our experiments as follows.

- **PostgreSQL** (pos, 1996) employs a default 1D histogram-based estimation method to analyze the distribution of individual columns.

- **DeepDB**[4] (Hilprecht et al., 2019) is a pure data-driven method, which learns the joint probability distribution of the underlying data using a Sum-Product-Network (SPN).

- **ALECE**[5] (Li et al., 2023b) is an attention-based regression method, which captures the relations between queries and data.

- **DDUp**[6] (Kurmanji & Triantafillou, 2023) is a two-stage approach, which first conducts loss-based out-of-distribution (OOD) detection and then uses knowledge distillation for model updates. DDUp utilizes NeuroCard as its base model for cardinality estimation and employs DBest++ for approximate query processing.

- **Fine-tune** (Kurmanji & Triantafillou, 2023) is based on DDUp's pipeline, with knowledge distillation being replaced by fine-tuning for model updates.

- **DBest++**[7] (Ma et al., 2021) utilizes mixture density networks (MDNs) to learn the probability density function of data and predict the results of the queries. We use DBest++FT to denote the approach that updates only frequency tables (FTs), whereas DBest++FT represents updating both FTs and MDNs to reflect its resemblance to fine-tuning.

- **KNN** is a distance-based method that classifies a data point based on the majority vote of its nearest neighbors.

- **RandomForest** constructs multiple decision trees using majority voting for classification and averaging their predictions for regression tasks.

- **MLP** is a fully connected feedforward neural network trained with stochastic gradient descent.

- **XGBoost**[8] (Chen & Guestrin, 2016) is an optimized gradient-boosting framework that builds an ensemble of decision trees sequentially.

- **LightGBM**[9] (Ke et al., 2017) is a gradient-boosting algorithm with a leaf-wise tree growth strategy.

---

[4] https://github.com/DataManagementLab/deepdb-public
[5] https://github.com/pfl-cs/ALECE
[6] https://github.com/meghdadk/DDUp
[7] https://github.com/qingzma/DBEstClient
[8] https://github.com/dmlc/xgboost
[9] https://github.com/microsoft/LightGBM

- **CatBoost**[10] (Prokhorenkova et al., 2018) is a boosting method optimized for handling categorical variables via ordered boosting and efficient one-hot encoding alternatives.

- **Type-LDD**[11] (Yu et al., 2023) is a drift-aware classifier via knowledge distillation.

- **SVR** maps inputs into a high-dimensional space and finds an optimal hyperplane that minimizes the error within a defined margin.

- **DecisionTree** recursively splits data into branches based on feature values, forming a hierarchical structure of decision nodes and leaf nodes.

- **GradientBoosting** trains weak decision trees iteratively, minimizing the residual error of the previous weak learners.

- **AutoGluon**[12] (Erickson et al., 2020) is an AutoML framework that employs multi-layer stack ensembling to combine diverse models, simplifying hyperparameter tuning and model selection for classification and regression tasks.

### G.3. Downstream Applications

In data systems, system-internal tasks such as cardinality estimation, database tuning, and transaction throughput measurement provide immediate ground-truth outcomes critical for query optimization, resource management, and system performance monitoring. However, obtaining ground truth for certain user-oriented tasks is more complex. In these cases, our context memory in FLAIR establishes a virtuous cycle of user feedback, where users can provide feedback on the model's predictive outcomes. This feedback acts as a practical form of ground truth, facilitating continuous refinement of model performance on user-oriented tasks and enabling system customization.

In our study, we evaluate FLAIR across four critical tasks in data systems, spanning from internal system functions to user-oriented activities. Detailed descriptions of each task and its setting are provided below.

**Cardinality Estimation** (CE) estimates the number of rows a query returns, aiding query planners in optimizing execution plans. We demonstrate FLAIR's in-context adaptation process using the cardinality estimation task as an example, as illustrated in Figure 9. In the cardinality estimation experiment, data-driven approaches such as DeepDB and DDUp are configured with the database data after executing all statements from the training portion of the workload, reflecting a real-world system scenario as described in (Li et al., 2023b). DeepDB is not compared on STATS as it only supports PK-FK joins. For FLAIR, the queue size $\varrho$ is set to 80, unless specified otherwise.

**Approximate Query Processing** (AQP) quickly delivers approximate results from large datasets by balancing accuracy with computational efficiency. In our evaluation of the AQP task, we adopt the same query schema used in prior works (Kurmanji & Triantafillou, 2023; Ma et al., 2021). Specifically, the test queries included 100 instances of SUM and AVG functions across various relations in the IMDB dataset. Following (Kurmanji & Triantafillou, 2023), the queries are randomly generated by selecting a lower and an upper bound for range filters and uniformly selecting a categorical column for the equality filter, providing a consistent and controlled testing environment. We instantiate the TFM for AQP tasks based on word embeddings to generate the task vector, following a methodology similar to DBest++. All methods use identical samples from the original dataset to ensure fairness in model building.

**In-database Data Analytics** involves data classification tasks and data regression tasks executed within the database engine, delivering insights directly from the data source. (1) **Data classification** boosts business intelligence by using categorical attributes to categorize tuples, such as product types and transaction statuses, supporting analytics in data systems. (2) **Data regression** predicts continuous outcomes, enhancing predictive analytics and decision-making on platforms like Oracle (Helskyaho et al., 2021) and Microsoft SQL Server (MacLennan et al., 2011; Harinath et al., 2008).

### G.4. Implementation Details

FLAIR is implemented in Python with Pytorch 2.0.1. In our experiments, we employ standard baselines such as KNN, MLP, and RandomForest from scikit-learn. Other baseline methods are implemented using their open-source packages or the source code provided by the respective researchers, which strictly adhere to the recommended configurations and settings.

---

[10] https://github.com/catboost/catboost
[11] https://github.com/liaub/Type-LDD
[12] https://github.com/autogluon/autogluon

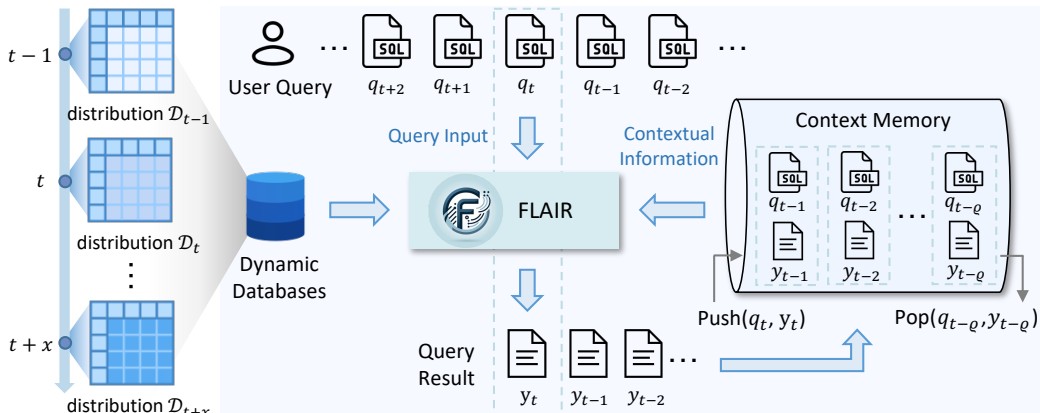

Figure 9: In-context adaptation for cardinality estimation.

The experiments involving PostgreSQL are conducted on PostgreSQL 13.1. All the experiments are conducted on a server with a Xeon(R) Silver 4214R CPU @ 2.40GHz (12 cores), 128G memory, and a GeForce RTX 3090 with CUDA 11.8. The OS is Ubuntu 20.04 with Linux kernel 5.4.0-72.

### G.5. Evaluation Metrics

We employ a comprehensive set of metrics to evaluate both the effectiveness and efficiency of our FLAIR across various downstream tasks, categorized into effectiveness metrics and efficiency metrics.

**Effectiveness Metrics.** For the CE task, we report accuracy by the widely recognized metrics Q-error and P-error. Q-error gauges the accuracy of estimated query cardinalities by measuring the discrepancy between the estimated cardinalities $c_{est}$ and the ground-truth cardinalities $c_{gt}$, as defined in Eq. 23. We report the geometric mean of the Q-error (GMQ) as (Li et al., 2022; Dutt et al., 2019) along with Q-error across various quantiles, with particular emphasis on the tail performance. P-error measures the gap between the optimal query plan $p_{opt}$, which uses the actual cardinality $c_{gt}$, and the plan $p_{est}$ derived using the estimated cardinality, as Eq. 24. It is quantified using a cost function $F_{cost}$, for which we adopt the default setting in PostgreSQL.

$$\text{Q-error} = \frac{\max(c_{est}, c_{gt})}{\min(c_{est}, c_{gt})} \in [1, +\infty) \tag{23}$$

$$\text{P-error} = \frac{F_{cost}(p_{est}, c_{est})}{F_{cost}(p_{opt}, c_{gt})} \in [1, +\infty) \tag{24}$$

For AQP task, we use mean relative error (MRE) as Eq. 25, which is widely utilized in previous related works (Ma & Triantafillou, 2019; Kurmanji & Triantafillou, 2023; Kurmanji et al., 2024) to evaluate the accuracy of query approximations for SUM and AVG aggregates.

$$\text{MRE} = \sum_{i=1}^{N} \frac{|c_{est}^i, c_{gt}^i|}{c_{gt}^i} \times 100 \tag{25}$$

For in-database data analytics, we apply accuracy and F1 score for data classification, both metrics range from 0 to 1, with higher values indicating better model performance. In data regression, we utilize mean squared error (MSE) and the coefficient of determination ($R^2$), where MSE ranges from 0 to infinity and $R^2$ ranges from 0 to 1. A lower MSE signifies a more accurate regression model, while a higher $R^2$ indicates better performance.

**Efficiency Metrics.** We assess FLAIR's efficiency by examining storage overhead, building time, inference time, and adaptation time. Specifically, storage overhead gauges the memory requirement of a method. Building time measures the necessary offline training duration, while inference time indicates the average time per input instance for estimation, crucial for real-time applications. Lastly, adaptation time reflects how quickly the model can adjust to concept drift. Additionally,

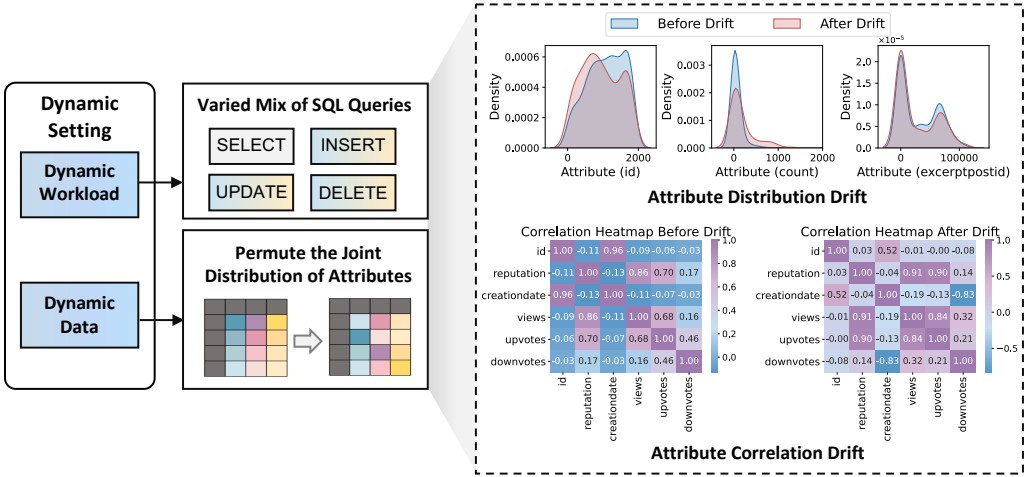

Figure 10: Overview of dynamic settings, illustrated by distribution discrepancies confirmed by Kolmogorov-Smirnov test p-values below 0.01 before and after concept drift.

we evaluate the efficiency of FLAIR within PostgreSQL by testing its query execution latency, which directly connects to the query optimizer and objectively shows how our method can enhance DBMS query performance.

### G.6. Dynamic Settings and Data Drift

In our study, we explore a dynamic data system marked by variations in both workload and data, which is illustrated in Figure 10. To emulate a real system environment, we introduce significant data drift after training and before testing. This involves sorting each column to alter the joint distribution of attributes and then performing random sampling from this permuted dataset. The impact of these manipulations on data distribution and attribute correlations is visually depicted through histograms and heat maps in Figure 10, showcasing the data characteristics before and after experiencing data drift. This dynamic scenario comprehensively mirrors real-world database operations where frequent insert, delete, and update actions induce gradual changes in data distribution. Over time, these incremental modifications accumulate, resulting in more pronounced shifts in data structures and inter-attribute relationships. To rigorously assess the robustness of our approach, we design two scenarios based on the extent and nature of the changes.

- **Mild Drift**: We randomly select 50% of records from the database and independently permute their column values, altering data distribution and inter-column correlations.

- **Severe Drift**: We randomly select 60% of records, independently permuting their columns, and performing random insertions, deletions, and updates, which affects 10% of the total data (keeping the total data size constant).

## H. Supplementary Experimental Results on In-database Data Analytics.

### H.1. Data Classification

We conduct sentiment analysis (Maas et al., 2011) on IMDB, which is a prevalent binary classification task. We allocate 50% of the original data as the training set, and following prior setups, induced data drift on the remaining data. We designate 20% of the post-drift data as the update set and the remaining post-drift data as the test set. For models that support incremental updates, such as XGBoost, LightGBM, CatBoost, and MLP, we incrementally update the models initially trained on the training set using the update set, while others are retrained on the update set. Finally, we evaluate all models on the test set to measure their effectiveness in adapting to data drift, as summarized in Table 3. The mean time represents the total execution time, integrating building, adaptation, and inference time averaged across two drift scenarios. Our FLAIR distinctly showcases its robustness and adaptability in handling concept drift, resulting in superior performance across both mild and severe drift scenarios. Furthermore, FLAIR achieves this high accuracy while maintaining impressive

computational efficiency compared with AutoGluon, making it exceptionally suited for practical dynamic environments where both performance and speed are crucial.

Table 3: Performance of data classification on concept drift.

| Category | Method | Mild Drift | | Severe Drift | | Mean |
|---|---|---|---|---|---|---|
| | | Acc↑ | F1↑ | Acc↑ | F1↑ | Time(s)↓ |
| Classical Non-linear Classifier | KNN | 0.795 | 0.591 | 0.586 | 0.379 | **1.469** |
| | RandomForest | 0.921 | 0.891 | 0.621 | 0.334 | 8.893 |
| | MLP | 0.852 | 0.585 | 0.676 | **0.496** | 15.798 |
| GBDT Classifier | XGBoost | 0.905 | 0.896 | 0.596 | 0.385 | 55.681 |
| | LightGBM | 0.870 | 0.727 | 0.595 | 0.377 | 16.765 |
| | CatBoost | 0.918 | 0.906 | 0.607 | 0.368 | 14.077 |
| AutoML System | AutoGluon | **0.936** | 0.908 | **0.679** | 0.441 | 85.183 |
| Ours | FLAIR | **0.932** | **0.920** | **0.826** | **0.632** | **8.377** |

## H.2. Data Regression

Table 4 offers a comprehensive comparison of representative regression methods in the context of concept drift, focusing on movie rating prediction (IMD, 2024), a scenario typically characterized by evolving concepts. FLAIR excels in both mild and severe drift scenarios, maintaining consistent performance across MSE and $R^2$ metrics while demonstrating comparable efficiency. While AutoGluon delivers the best results under mild drift conditions, its performance noticeably declines under severe drift and requires more than $40\times$ computational time compared to FLAIR.

Table 4: Performance of data regression on concept drift.

| Category | Method | Mild Drift | | Severe Drift | | Mean |
|---|---|---|---|---|---|---|
| | | MSE↓ | $R^2$↑ | MSE↓ | $R^2$↑ | Time(s)↓ |
| Classical Method | SVR | 0.591 | 0.230 | 0.691 | 0.210 | **0.081** |
| | MLP | 8.762 | -10.418 | 28.355 | -49.003 | 10.259 |
| Tree-based Method | DecisionTree | 0.557 | 0.231 | 0.652 | 0.198 | **0.068** |
| | RandomForest | 0.315 | 0.570 | 0.458 | 0.475 | 0.942 |
| | GradientBoosting | 0.325 | 0.577 | **0.396** | 0.487 | 0.355 |
| AutoML System | AutoGluon | **0.267** | **0.682** | 0.399 | **0.632** | 27.438 |
| Ours | FLAIR | **0.271** | **0.647** | **0.388** | **0.647** | 0.681 |

# I. Ablation Study

## I.1. Effects of Queue Size in Context Memory

We further analyze the sensitivity of FLAIR to the critical hyperparameter $\varrho$, the size of queues in context memory, across various benchmarks and dynamic scenarios, as depicted in Figure 11. The results confirm that increasing the queue size contributes to performance enhancements without escalating system latency, owing to embedding cache optimization. Initially, performance improves significantly with an increase in queue size but eventually plateaus, indicating diminishing returns. Notably, an oversized queue size may introduce information redundancy, potentially leading to a performance decline. For instance, increasing the queue size to 100 results in minor deterioration in the STATS benchmark's mild drift scenario. In summary, the optimal queue size $\varrho$ should be tailored based on the complexity of the data to balance performance gains against the risk of redundancy, in order to optimize the model's efficacy in dynamic environments.

## I.2. Effects of Histogram Granularity

To evaluate the impact of histogram binning granularity in the TFM, we conduct a sensitivity analysis by varying the number of bins $\delta$ used in data encoding. Specifically, we test $\delta \in \{10, 20, 40, 60, 80\}$ across four scenarios involving two

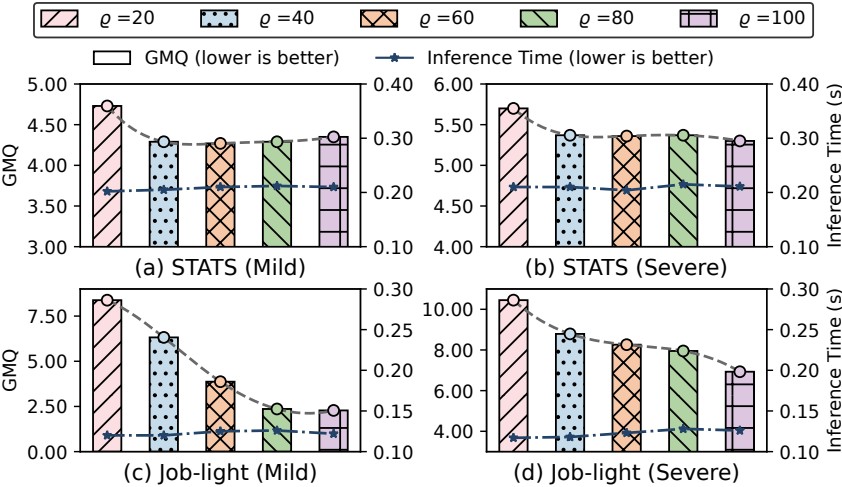

Figure 11: Sensitivity analysis of the queue size $\varrho$.

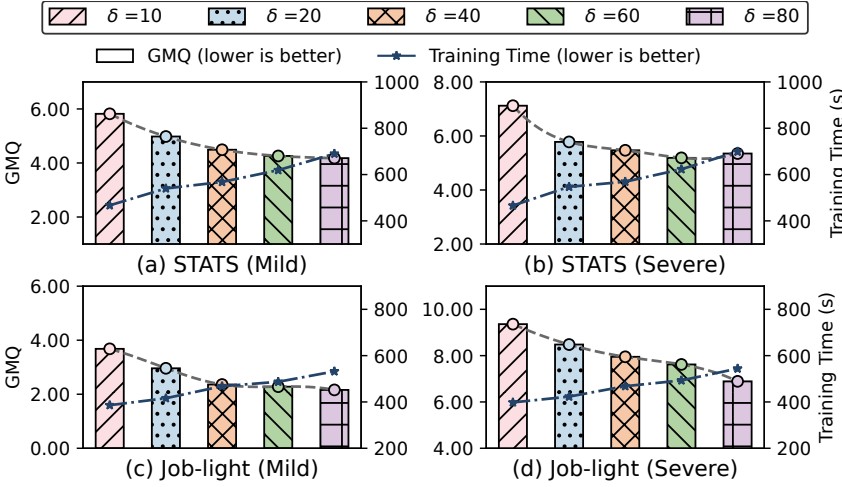

Figure 12: Sensitivity analysis of the bin number $\delta$.

benchmarks under mild and severe dynamic scenarios. As shown in Figure 12, decreasing the bin number generally improves performance in terms of GMQ but increases the training time. Notably, $\delta = 40$ achieves a favorable trade-off between encoding fidelity and computational efficiency, yielding significant reductions in GMQ without incurring excessive training time.

### I.3. Effects of User Feedback

To delve into the adaptability of FLAIR in user-oriented tasks, we evaluate how varying proportions of user feedback data $\rho$ within queues affect model performance. We use drifted data with ground-truth outputs to simulate user-customized feedback data, assessing the model's conformity to user-specific requirements. Specifically, the queues comprise a certain proportion of user feedback data combined with the model's recent input-output pairs. We maintain the queue size at 80 and vary the proportion of user feedback data. The results in Figure 13, demonstrate that increasing the proportion $\rho$ within a fixed queue size significantly enhances model performance, confirming the model's ability to be customized by users. To further explore the impact of integrating recent model interactions into the queue on performance, we conduct comparative experiments using only user feedback data. We observe that mixed queues outperform those containing solely user feedback. Additionally, integrating recent model data mitigates performance decline as the proportion $\rho$ of user feedback decreases.

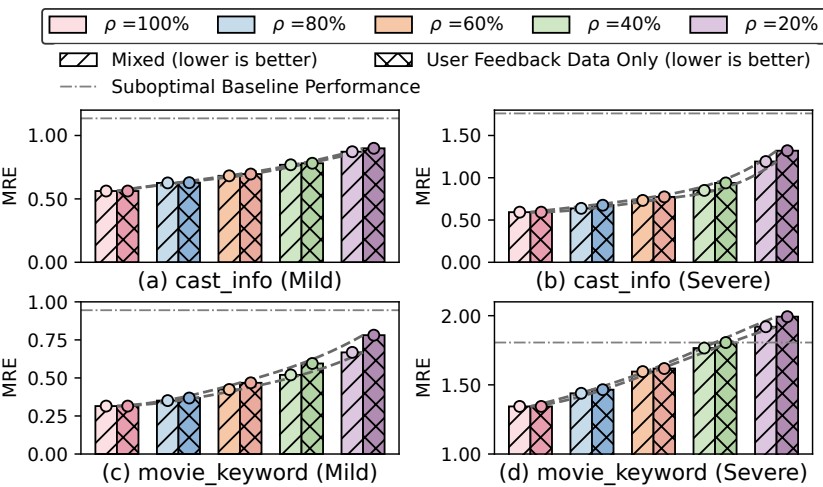

Figure 13: Sensitivity analysis of the user feedback $\rho$.

Still, we advise against setting $\rho$ too low due to the risk of introducing noise. It is noteworthy that FLAIR surpasses the suboptimal model DDUp at most times even with very low $\rho$, underscoring FLAIR's capability in user-oriented applications.

