# OpenReview forum: "In-Context Adaptation to Concept Drift for Learned Database Operations"
_ICML.cc/2025/Conference — ICML 2025 poster_

### Official Review · Reviewer_C5eR · 2025-03-10

**Overall Recommendation:** 3

**Summary:**

The paper introduces an online adaptation framework to tackle concept drift in learned database operations. The topic is related to the application of machine learning. The proposed FLAIR leverages in-context adaptation through dynamic context memory and Bayesian meta-training, enabling models to adjust to evolving data distributions without retraining. The framework comprises a Task Featurization Module for standardizing inputs and a Dynamic Decision Engine pre-trained on synthetic data to generalize across tasks. Experiments on benchmarks like STATS and JOB-light demonstrate FLAIR’s competitive performance in severe concept drift.

**Claims And Evidence:**

Yes

**Essential References Not Discussed:**

NA

**Experimental Designs Or Analyses:**

Yes

**Methods And Evaluation Criteria:**

Yes

**Other Comments Or Suggestions:**

NA

**Other Strengths And Weaknesses:**

# Strengths
1.	The topic of concept drift is related to the machine learning field.
2.	Solving the application of database with concept drift has the potential to boost real-world applications.
3.	The proposed FLAIR is evaluated on several datasets against other baselines.

# Weaknesses
1.	In the proposed method, the Bayesian meta-training step relies on synthetic datasets sampled from predefined priors. Although it seems to improve the performance, the synthetic data may fail to capture real-world distribution complexities, leading to gaps in generalization. The paper lacks validation of how synthetic priors align with actual drift patterns encountered in dynamic databases.
2.	Besides, the current framework assumes immediate availability of execution results for context memory updates. In the real system, complex queries or distributed systems may introduce latency or partial feedback, limiting the framework’s applicability in real-time or large-scale environments. Hence, the authors should consider more complex experimental scenarios.
3.	The experiments focus on moderate-scale datasets There is no analysis on the computational overhead, memory footprint, or latency when applied to high-velocity data streams or petabyte-scale databases.


# Remark
To me, this paper would be a better fit for the internation conferences on database, where the audience would benefit more than the machine learning field.

**Questions For Authors:**

NA

**Relation To Broader Scientific Literature:**

The paper introduces an online adaptation framework to tackle concept drift in learned database operations. The topic is related to the application of machine learning. The proposed FLAIR leverages in-context adaptation through dynamic context memory and Bayesian meta-training, enabling models to adjust to evolving data distributions without retraining. The framework comprises a Task Featurization Module for standardizing inputs and a Dynamic Decision Engine pre-trained on synthetic data to generalize across tasks. Experiments on benchmarks like STATS and JOB-light demonstrate FLAIR’s competitive performance in severe concept drift.

**Theoretical Claims:**

Yes

---

> ### Author Rebuttal · Authors · 2025-04-01
>
> Dear Reviewer C5eR,
>
> We sincerely thank you for the constructive feedback that helps improve our work. We address the concerns you raised below. The full new results are provided in: https://anonymous.4open.science/r/ICML25-7F63/ICML25.pdf
>
> **[Q1-Synthetic Priors]** We clarify that FLAIR's DDE is meta-trained using tasks from BNNs and SCMs, which model complex dependencies and causal structures fundamental in the data stored in real-world databases.
> The priors span diverse functional forms reflecting causal relationships in real query-data-output relations and are trained in a fully Bayesian manner over a wide hyperparameter space broader than any single-task point estimate.
> We also adopt Occam’s razor bias as [1], favoring lower-parameter models, which aligns with Bayesian and cognitive principles.
>
> We validate the generalizability of the synthetic priors empirically: We meta-train FLAIR once and for all and test it across 4 core analytics tasks (i.e., CE, AQP, classification), reflecting distinct data-query patterns and dynamics.
> The strong and consistent performance across all these tasks, even superior to the strong baseline fine-tuning on real-world data, confirms the generalizability of our synthetic priors.
>
> As suggested, we further validate this with new experiments simulating insert/delete/update-heavy drifts. Results below (details in Table A) show robust and optimal performance, especially in update-heavy (UH) cases.
> This confirms the alignment between our priors and actual drift patterns.
> |||Mild|||Severe||
> |-|-|-|-|-|-|-|
> |Data|IH|DH|UH|IH|DH|UH|
> |STATS|2.85|2.96|3.76|3.35|3.26|3.96|
> |Job-light|1.68|1.59|1.92|6.26|6.58|7.21|
>
> **[Q2-Latency/Partial Feedback]** We agree with you that the immediate availability of execution results may not always hold.
> To address this, as suggested, we extend our evaluation to consider more complex scenarios and design two settings:
>
> Delayed feedback: context memory updates every k steps (5% and 10% of its size) rather than in real time.
>
> Partial feedback: 5% and 10% of context pairs are randomly dropped to mimic missing or partial feedback.
>
> The results below (details in Table G) show that: (1) partial feedback performs slightly better, suggesting recency of execution results matters more than completeness. (2) FLAIR is resilient to both cases, which confirms its applicability in real systems.
> |Data|Drift|Delay 5%|Delay 10%|Partial 5%|Partial 10%|FLAIR|
> |-|-|-|-|-|-|-|
> |STATS|Mild|4.79|5.32|4.52|4.88|4.49|
> ||Severe|5.62|5.92|5.53|5.61|5.47|
> |Job-light|Mild|2.45|2.56|2.40|2.49|2.36|
> ||Severe|8.10|8.26|8.06|8.12|7.95|
>
> **[Q3-Larger Datasets]** We share your view that scalability is a key consideration beyond effectiveness. As discussed in Appx G, FLAIR is designed with scalability in mind with complexity linear scaling w.r.t. input and context size.
> Its runtime involves only a feedforward pass over compact context memory for adaptation, without backward gradient updates.
>
> We evaluated FLAIR on real and widely accepted benchmarks STATS and JOB-light, which show low inference latency (Sec 4.3) and stability with context memory scaling (Fig 11, Appx J1). To further evaluate the scalability of our FLAIR, we add new experiments on a larger dataset TPC-H [2](10GB, 86M records, complex many-to-many joins).
> Results in Table H show that FLAIR outperforms baselines in all cases, with 3.5% and 7.2% gains under mild and severe drift.
>
> Across all benchmarks, FLAIR maintains low latency(\~10 ms/query), compact memory footprint(~5 MB), and stable storage overhead, showing its suitability for high-throughput, large-scale systems.
> |Data|Throughput(q/s)|Memory Footprint(MB)|Latency(ms/q)|Storage Overhead(MB)|
> |-|-|-|-|-|
> |STATS|109.17|4.97|9.16|47.68|
> |Job-light|121.21|5.02|8.25|47.23|
> |TPC-H|93.28|5.11|10.82|47.72|
>
> **[Remark]** We clarify that our work sits at the intersection of machine learning and data systems.
> The key insights of our work lie in addressing two fundamental ML challenges: how to enable on-the-fly adaptation without retraining and how to achieve context-aware prediction, as stated in lines 036–042.
> These challenges are central to real-time analytics on dynamic structured data for high stacks applications such as healthcare and stock trading.
> Thus, we focus on typical structured data analytics tasks, addressing these challenges with contributions grounded in ML, i.e., in-context adaptation and Bayesian meta-training, achieving promising performance.
> We believe this work contributes to the broader ML community and will resonate with a wide ML audience working on concept drift, structured data analytics, and real-time systems, complementing ongoing efforts in these areas.
>
> We are grateful for the chance to discuss our work's potential and wish to thank you again for your valuable input.
> We hope to have addressed your concerns and would highly appreciate your consideration for re-evaluating your initial rating.
>
> [1]Tabpfn. ICLR2023
>
> [2]https://www.tpc.org/

---

### Official Review · Reviewer_5u5N · 2025-03-12

**Overall Recommendation:** 3

**Summary:**

This paper addresses the challenge of concept drift in database operations. Its primary contribution is the introduction of an in-context adaptation framework to tackle this issue. The proposed method, FLAIR, comprises two essential components: a Task Featurization Module and a Dynamic Decision Engine. The effectiveness of FLAIR is demonstrated through both theoretical analysis and experimental evaluation.

**Claims And Evidence:**

1. The intuition behind the in-context adaptation framework is outlined in Theorems 3.1 and 3.2, which I find reasonable.

2. The experimental results demonstrate that the proposed method performs well in real-world applications. However, I am concerned that the baseline methods used for comparison are somewhat outdated and may not specifically address the distribution shift problem. I would expect a comparison with more recent approaches designed to tackle distribution shift.

**Essential References Not Discussed:**

N/A

**Experimental Designs Or Analyses:**

In Figure 7, the authors only compare with some very classical methods. I am confused with the information conveyed in this figure.

**Methods And Evaluation Criteria:**

I am not well-versed in database operation problems, but the evaluation criteria appears reasonable to me. However, I am confused about how the authors define the mild drift and severe drift.

**Other Comments Or Suggestions:**

N/A

**Other Strengths And Weaknesses:**

N/A

**Questions For Authors:**

N/A

**Relation To Broader Scientific Literature:**

This paper introduces the concept of "concept drift." However, from my perspective, it appears to be no different from the distribution shift problem. Further clarification is needed.

**Theoretical Claims:**

I do not consider this paper to have theoretical innovation. However, since its primary focus is on experimental analysis, I believe the current theoretical discussion is sufficient to illustrate and validate the authors' intuition. I don't find any issues in the proof.

---

> ### Author Rebuttal · Authors · 2025-04-01
>
> Dear Reviewer 5u5N,
>
> We thank you for your recognition of our work and your insightful comments. Below, we provide detailed responses to the specific concerns you raised. Full tables/figures are in: https://anonymous.4open.science/r/ICML25-7F63/ICML25.pdf
>
> > expect a comparison with more recent approaches.
>
> As suggested, we have added recent baselines:
> SOLID [1], a context-aware fine-tuning adaptation method, and DtACI [2], an adaptive conformal inference (ACI)–based fine-tuning method.
> Results below (details in Table F) show that FLAIR outperforms both under drift.
> This is because SOLID’s residual-based detection and DtACI’s ACI mechanism miss subtle, continuous query-driven drift in databases, reducing accuracy, especially in tail cases.
> In contrast, FLAIR’s in-context adaptation ensures timely and better adaptation to continuous drift.
> |Data|Method|Mild|Severe|
> |-|-|-|-|
> |STATS|SOLID|5.12|5.56|
> ||DtACI|5.48|6.86|
> ||FLAIR|4.49|5.47|
> |Job-light|SOLID|2.41|8.36|
> ||DtACI|2.54|8.12|
> ||FLAIR|2.36|7.95|
>
> > how the authors define the mild drift and severe drift？
>
> Thank you for pointing this out. We clarify the drift setting below:
>
> Mild Drift: randomly select 50% of records from the database and independently permute their column values, altering data distribution and inter-column correlations.
>
> Severe Drift: randomly select 60% of records, independently permuting their columns, and performing random insertions, deletions, and updates, which affects 10% of the total data (keeping the total data size constant).
>
> This setting follows the recent work [3]. We will refine the descriptions in Figure 10 and Appx H6 for clarity.
>
> > confused with information conveyed in Figure 7
>
> We appreciate your valuable feedback. We selected classical methods in Figure 7 for their interpretable decision boundaries, which provide an intuitive understanding of model behavior under concept drift.
>
> To enhance comparison, we have added a recent method Type-LDD [4], a drift-aware classifier via knowledge distillation, in Figure C. While Type-LDD surpasses classical methods, FLAIR outperforms it due to the Type-LDD’s delayed adaptation from its detect-then-adapt strategy.
>
> > the concept of 'concept drift' appears to be no different from the distribution shift problem
>
> We agree that clearly distinguishing our core focus concept drift in databases from the broader notion of distribution shift is important, and we would like to further clarify this as suggested.
> Distribution shift typically refers to a mismatch between training and test data distributions, often without inherent temporal evolution.
> In contrast, concept drift in our work refers to the ongoing, temporal evolution of data and queries in operational database settings. In databases, this form emerges naturally: runtime queries continuously trigger insert/delete/update operations, incurring cumulative changes in both data and query. These drifts are ongoing and unpredictable, requiring models to adapt in real time rather than being trained repeatedly with new data distributions.
>
> To clarify this clearly, we refine our problem definition as follows:
>
> **Definition** (Concept Drift in Databases) Let $\mathbf{d}_t$ denote the underlying data of a database at time t, and $\mathbf{q}_t$ denote a user query at time t. Given the data-query pair $(\mathbf{d}_t,\mathbf{q}_t)$, let $\mathbf{y}_t$ represent the corresponding prediction output (e.g., row counts in cardinality estimation). Concept drift occurs at time t if the joint distribution of queries, data, and predictions changes, i.e.,
>
> $P_t(\mathbf{q},\mathbf{d},\mathbf{y})\ne P_{t+1}(\mathbf{q},\mathbf{d},\mathbf{y})$.
>
> Here, drift can arise from two distinct but interrelated sources:
>
> (1) Query drift, from evolving user behavior.
>
> (2) Data drift, caused by frequent insert/delete/update operations changing underlying data distributions.
>
> Notably, changing data not only changes the marginal distribution $P(\mathbf{d})$, but also affects the conditional distribution $P(\mathbf{y}|\mathbf{q},\mathbf{d})$, i.e., the same queries may yield different outputs over time.
> This suggests that concept drift in databases involves shifts in the joint distribution of queries, data, and predictions, and their interaction.
>
> We'll further clarify this distinction and update our problem formulation accordingly.
>
> We hope our responses above have sufficiently addressed your concerns and can improve your evaluation of our work.
>
> [1] Calibration of Time-Series Forecasting- Detecting and Adapting Context-Driven Distribution Shift. KDD 2024
>
> [2] Conformal Inference for Online Prediction with Arbitrary Distribution Shifts. JMLR 2024
>
> [3] Detect, Distill and Update: Learned DB Systems Facing Out of Distribution Data. SIGMOD 2023
>
> [4] Type-LDD: A Type-Driven Lite Concept Drift Detector for Data Streams. TKDE 2024

---

### Official Review · Reviewer_uD1S · 2025-03-12

**Overall Recommendation:** 4

**Summary:**

This paper focuses on the issue of concept drift in dynamic database environments, which is an interesting and challenging research problem. To address this problem effectively, an online adaptation framework, FLAIR, has been developed. Sufficient experiment and analysis show the performance of the proposed method. I have some suggestions for further improvement.

**Claims And Evidence:**

Yes, sufficient experiment on different datasets and baselines have been conducted to evaluate the performance of the proposed method, and detailed analysis of the results also verify the efficiency.

**Essential References Not Discussed:**

The key contribution is an online adaptation framework, called in-context adaptation for learned database operations under concept drift, I have seen many database related reference has been listed, however, the related work about concept drift learning is insufficient, I suggest adding more reference about concept with detailed discussion.

**Experimental Designs Or Analyses:**

I have reviewed the experimental designs and analysis, two benchmark datasets have been chosen for the experiment, I still suggest the author add more datasets for model evaluation.
Besides, the parameter setting of the proposed method and all the baselines should be introduced in this paper, and a experiment for parameter analysis is needed.

**Methods And Evaluation Criteria:**

The proposed method has been clearly introduced and evaluated, and the evaluation criteria have been chosen appropriately, but I have some suggestion for this part, as shown below:

1.	In the dynamic database, the data may have incremental updates, leading to concept drift. However, the users’ queries may also change as time goes on. So, does this work consider this situation when both the data and query change?
2.	It seems the proposed method does not have the drift detection process, so, how to identify the impact of concept drift on model performance, which is very important for learning adaptation.
3.	For the proposed FLAIR, a meta-trained part is embedded, which has been frozen, so how to ensure the efficiency of this part when concept drift occurs? I understand you have extracted features from data and query, but the adaptation process of the meta-trained model should be explained clearly in Figure 3. Besides, training and fine-tuning are common solutions for concept drift adaptation, and the proposed method dynamically adapts to new concepts guided by the context memory during inference, have you compared the performance of these three learning strategies?
4.	The runtime complexity analysis of the proposed is required.

**Other Comments Or Suggestions:**

Based on the comparison of learning efficiency in Figure 4, the runtime complexity analysis of the proposed is required.

**Other Strengths And Weaknesses:**

1.	Is this research work in the supervised learning setting? I know the definition of concept drift is $P_{t}(y|x) \neq P_{t+1}(y|x)$ is based on the setting of supervised learning, so, does the data in the database with different category (at each time point) have the ground truth (label)?
2.	For definition 2.1, I think this definition should show the difference of concept drift that occurs in dynamic data stream and dynamic database. Please explain and define it again.

**Questions For Authors:**

I suggest the author clarify the setting of this research work, supervised learning (regression or classification? or both of them?), and the experiment of regression and classification should be analyzed separately.

**Relation To Broader Scientific Literature:**

This paper address the issue of concept drift in dynamic database environments, which attracts high attention in recent years. Concept drift learning is a challenge research topic in data mining, concept drift occurs in dynamic data and many previous studies develops drift detection and adaptation method. This paper develops method for concept drift adaptation in the scenario of dynamic database, which is an interesting topic in this area, and sufficient comparison experiment verified the performance.

**Theoretical Claims:**

The author gives the theoretical analysis of model generalization error bound analysis with sufficient proof and analysis. However, the target of the proposed method is to handle concept drift in a dynamic database environment, I have not seen the author analyze bound under concept drift respectively. That is to say, the theoretical analysis should embed the term of concept drift that occurs dynamically.

---

> ### Author Rebuttal · Authors · 2025-04-01
>
> Dear Reviewer uD1S,
>
> We sincerely thank you for your positive review and insightful comments. We address your concerns below and attach new results here: https://anonymous.4open.science/r/ICML25-7F63/ICML25.pdf
>
> > Q1: does the work consider both data and query change?
>
> Yes, we considered both data and query change in Appx H6 (line 1287). We evaluated the query change of SELECT/UPDATE/INSERT/DELETE with varied ratios, and among them, UPDATE/INSERT/DELETE incur data change.
>
> We further evaluate our pretrained model on insert/delete/update-heavy (IH/DH/UD) query changes. Results below (details in Table A) show that FLAIR adapts consistently well to both query and data changes across different settings.
> |||Mild|||Severe||
> |-|-|-|-|-|-|-|
> |Data|IH|DH|UH|IH|DH|UH|
> |STATS|2.85|2.96|3.76|3.35|3.26|3.96|
> |Job-light|1.68|1.59|1.92|6.26|6.58|7.21|
>
> > Q2: how to identify the impact of concept drift?
>
> Unlike reactive detect-then-adapt methods, FLAIR uses the dynamic context memory to adapt on the fly via the dynamic decision engine, avoiding the detection overhead.
>
> Incremental drift tests (Figure 5 in Sec 4.4) and our new abrupt drift tests below (details in Table B) confirm that FLAIR achieves the best performance without explicit detection.
> |Data|PG|ALECE|DDUp|FT|FLAIR|
> |-|-|-|-|-|-|
> |STATS|176.38|12.63|6.91|5.75|4.15|
> |Job-light|19.41|16.24|6.65|6.25|3.26|
>
> > Q3: how to ensure efficiency of meta-trained part? have you compared common learning strategies?
>
> DDE is meta-trained only once to approximate $q_{\theta}(y|x,\mathcal{C})$ across diverse tasks. During inference, it remains frozen and adapts efficiently via a forward pass using context $\mathcal{C}$, built from recent query-output pairs, thereby enabling training-free adaptation.
>
> As suggested, we update Figure 3 for clarity (see Figure A) and add training from scratch (RT) besides the fine-tuning (FT) and distillation (KD) baselines (details in Table C). Results show that RT and FT underperform in adaptation to new concepts, as they fit outdated data, while FLAIR archives timely and better adaptation.
> |Data|Drift|RT|FT|FLAIR|
> |-|-|-|-|-|
> |STATS|Mild|4.97|5.35|4.49|
> ||Severe|5.59|5.02|5.47|
> |Job-light|Mild|3.25|2.45|2.36|
> ||Severe|8.21|8.09|7.95|
>
> > Q4: runtime complexity analysis
>
> FLAIR's initialization complexity is $O(\delta \sum n_i)$ with $O(N_v)$ for updating modified records. Query encoding cost $O(N_J+N_F)$ is negligible. TFM incurs $O(d_a\delta^2)$ (self-attention) and $O(d_a\delta)$ (cross-attention). DDE scales linearly as $O(d_a\varrho)$ with context size $\varrho$. Please see Appx G for more details.
>
> > theoretical analysis should embed concept drift
>
> We clarify that we model concept drift through insertions and deletions, which are key drivers of drift in databases, and derive worst-case bounds. As suggested, we will make the link to concept drift clearer in Sec 3.4.
>
> **[Additional Dataset]** We add TPC-H [1] (10GB, 86M records, 8 tables, 61 attributes). Experimental settings follow Appx H6.
> Results below show that FLAIR outperforms all the baselines, especially in tail and severe cases (details in Table D).
> |Drift|FT|PG|ALECE|DDUp|FLAIR|
> |-|-|-|-|-|-|
> |Mild|3.75|36.15|8.97|6.58|3.62|
> |Severe|6.11|88.75|38.05|9.65|5.67|
>
> **[Parameter Analysis]** As suggested, we clarify key parameter settings: the bin number is 40, context memory size is 80, the task encoder 8 attention layers with 8 heads, and DDE 12 layers with 4 heads. Detailed settings for FLAIR and baselines will be added to Appx H4.
>
> We add analysis on bin number $\delta$ (see Figure B), showing trade-off between accuracy and training time. Appx J provides more sensitivity analysis on context memory.
>
> **[Related Work]** As suggested, we expand related work with more references and discussion (see Table E).
> We group existing methods into:
> Lazy: retrain models after drift detection, high cost;
> Incremental: adapt gradually, respond slowly;
> Ensemble: maintain model pool to cover varied concepts, resource-heavy.
> In contrast, FLAIR proposes a new in-context adaptation paradigm, enabling timely adaptation without retraining.
>
> **[Setting Clarification]** We clarify that our work is in the supervised setting. For CE and AQP, labels come from query execution (Appx H1). For regression/classification, ground truth is available pre- and post-drift, with results in Appx J1/J2, respectively. We'll clarify this in Sec 4.5.
>
> **[Definition Refinement]** As suggested, we refine Definition 2.1 to distinguish drift in databases from drift in data streams. In databases, predictions depend on both query and data, and the data evolves due to query-driven operations. We thus refine the input into two distinct but interrelated parts: query and data, both can incur drift in databases. Due to space limit, please see our response to Reviewer 5u5N-Q4 for the updated definition.
>
> We hope our clarifications and new results have addressed your concerns and can improve your evaluation of our work.
>
> [1] https://www.tpc.org/

---

> > ### Comment · Reviewer_uD1S · 2025-04-05
> >
> > I have read the author's reply. Thank you very much. The author has explained and answered my comments in detail and provided experimental analysis to verify the effectiveness and innovation of the work. I will improve the score.

---

> > > ### Author Response · Authors · 2025-04-05
> > >
> > > Thank you very much for your invaluable feedback and for raising the score. Your insightful comments are truly inspiring and have greatly enhanced the quality of our work. We sincerely appreciate your time and effort.

---

### Decision · Program_Chairs · 2025-05-01

**Decision:**

Accept (poster)

**Comment:**

This paper proposes an online adaptation framework for learned database operations, introducing a novel in-context adaptation paradigm to address concept drift in dynamic database environments. The core task and methodology are developed within the context of databases. By leveraging immediate execution feedback and dynamic context memory, the proposed method avoids costly retraining and adapts at runtime through a task featurization module and a Bayesian meta-trained decision engine.

Initially, the paper received scores of 3, 3, and 2, with reviewers raising concerns about the realism of using synthetic priors for meta-training, the assumption of immediate feedback availability, the lack of comparison with stronger baselines for distribution shift, limited discussion of query drift or joint query-data drift, absence of explicit drift detection, and insufficient analysis of runtime complexity and scalability in large-scale systems.

Following the rebuttal, reviewers acknowledged that part of concerns were addressed and raised their scores to 4, 3, and 3.

The AC agrees that the paper contains sufficient values to be accepted but is slightly concerned by the clarity of the concepts and some technical details, the relationship with other methods, the potentials issue of the synthetic data usage, and the scope of the work. While the topic intersects with machine learning, Reviewer C5eR noted that the paper may have greater impact within the database community.